



# Evaluation of simulated $CO_2$ power plant plumes from six high-resolution atmospheric transport models

Dominik Brunner[1], Gerrit Kuhlmann[1], Stephan Henne[1], Erik Koene[1], Bastian Kern[2], Sebastian Wolff[2], Christiane Voigt[2,7], Patrick Jöckel[2], Christoph Kiemle[2], Anke Roiger[2], Alina Fiehn[2], Sven Krautwurst[3], Konstantin Gerilowski[3], Heinrich Bovensmann[3], Jakob Borchardt[3], Michal Galkowsi[4], Christoph Gerbig[4], Julia Marshall[4,2], Andrzej Klonecki[5], Pascal Prunet[5], Robert Hanfland[6,7], Margit Pattantyús-Ábrahám[6], Andrzej Wyszogrodzki[8], and Andreas Fix[2]

[1]Empa, Swiss Federal Laboratories for Materials Science and Technology, Dübendorf, Switzerland
[2]Deutsches Zentrum für Luft- und Raumfahrt (DLR), Institut für Physik der Atmosphäre, Oberpfaffenhofen, Germany
[3]Institute for Environmental Physics, University of Bremen, Bremen, Germany
[4]Max-Planck-Institute for Biogeochemistry, Jena, Germany
[5]SPASCIA, Ramonville Saint Agne, France
[6]Federal Office for Radiation Protection, Oberschleißheim, Germany
[7]Institute of Atmospheric Physics, Johannes Gutenberg University Mainz, Mainz, Germany
[8]Institute of Meteorology and Water Management - National Research Institute, Warsaw, Poland

**Correspondence:** Dominik Brunner (dominik.brunner@empa.ch)

**Abstract.** Global anthropogenic $CO_2$ sources are dominated by power plants and large industrial facilities. Quantifying the emissions of these point sources is therefore one of the main goals of the planned constellation of anthropogenic $CO_2$ monitoring satellites (CO2M) of the European Copernicus program. Atmospheric transport models may be used to study the capabilities of such satellites through observing system simulation experiments and to quantify emissions in an inverse modelling
5    framework. How realistically the $CO_2$ plumes of power plants can be simulated and how strongly the results may depend on model type and resolution, however, is not well known due to a lack of observations available for benchmarking. Here, we use the unique data set of aircraft in-situ and remote sensing observations collected during the CoMet measurement campaign downwind of the coal-fired power plants at Bełchatów in Poland and Jänschwalde in Germany in 2018 to evaluate the simulations of six different atmospheric transport models. The models include three Large-Eddy-Simulation (LES) models, two
10   mesoscale numerical weather prediction (NWP) models, and one Lagrangian particle dispersion model (LPDM) and cover a wide range of model resolutions from 200 m to 2 km horizontal grid spacing. At the time of the aircraft measurements between late morning and early afternoon, the simulated plumes were slightly (at Jänschwalde) to highly (at Bełchatów) turbulent, consistent with the observations, and extended over the whole depth of the atmospheric boundary layer (ABL, up to 1800 m a.s.l. in the case of Bełchatów). The stochastic nature of turbulent plumes puts fundamental limitations to a point-by-point
15   comparison between simulations and observations. Therefore, the evaluation focused on statistical properties such as plume amplitude and width as a function of distance from the source. LES and NWP models showed similar performance and sometimes remarkable agreement with the observations when operated at comparable resolution. A resolution of 1 km or better, however, appears to be necessary to realistically capture turbulent plume structures. At coarser resolution, the plumes disperse





too quickly especially in the near field (0-8 km from the source) and turbulent structures are increasingly smoothed out. Total
vertical columns are easier to simulate accurately than the vertical distribution of $CO_2$, since the latter is critically affected by
profiles of vertical stability, especially near the top of the ABL. Cross-sectional flux and integrated mass enhancement methods
applied to synthetic CO2M data generated from the model simulations with a random noise of 0.5 ppm – 1.0 ppm suggest
that emissions from a power plant like Bełchatów can be estimated with an accuracy of about 20% from single overpasses.
Estimates of the effective wind speed are a critical input for these methods. Wind speeds in the middle of the ABL appear to
be a good approximation for plumes in a well-mixed ABL as encountered during CoMet.

# 1  Introduction

According to a recent compilation of sectorial greenhouse gas emissions for the year 2018, approximately 34% of global anthropogenic $CO_2$ emissions are attributable to the energy sector and 24% to the industrial sector (Minx et al., 2021). Emissions
from these sectors primarily originate from power plants, industrial combustion plants and other large industrial facilities. The
concentrated plumes of these sources may be detectable from satellite observations (Nassar et al., 2017), which makes the quantification of these emissions an attractive target for observation-based $CO_2$ emission monitoring. Quantifying the emissions of
large point sources is indeed one of the main goals of the Anthropogenic $CO_2$ Emissions Monitoring and Verification Support
Capacity (CO2MVS) currently developed under Europe's Earth observation program Copernicus (Janssens-Maenhout et al.,
2020). This is not only important because of their large global share, but will also help us to better quantify the remaining more
dispersed emissions, which are not necessarily visible as plumes but rather as contributions to regional $CO_2$ enhancements.

Emissions from large combustion plants are often measured directly within the stacks, especially in economically more
developed countries, but these numbers are neither always readily and publicly available or only with large delays, nor is a
complete global record of power plant emissions realistically available in the near future. One of the main goals of the planned
European Copernicus Anthropogenic Carbon Dioxide Monitoring satellite mission (CO2M) is therefore to quantify the $CO_2$
emissions of large point sources globally by providing images of total column dry-air mole fractions ($XCO_2$) at a spatial
resolution of about 2 km x 2 km over a 250 km wide swath (Janssens-Maenhout et al., 2020).

A growing body of scientific literature has demonstrated the feasibility of quantifying $CO_2$ emissions from power plants
using satellite observations. These studies were either based on theoretical considerations combined with synthetically generated (simulated) $CO_2$ observations (Bovensmann et al., 2010; Kuhlmann et al., 2019; Strandgren et al., 2020; Kuhlmann
et al., 2021b) or on real observations from existing $CO_2$ satellites like OCO-2 (Nassar et al., 2017; Reuter et al., 2019; Nassar
et al., 2021; Hakkarainen et al., 2021; Kiemle et al., 2017; Chevallier et al., 2022) and hyperspectral imagers like PRISMA
(Cusworth et al., 2021).

Numerous methods have been proposed to quantify point source emissions from satellite observations using mass balance
considerations or by fitting a simulated plume to the observations (Krings et al., 2013; Varon et al., 2018; Beirle et al., 2019;
Kuhlmann et al., 2021b; Fioletov et al., 2015). Plume fitting methods often rely on Gaussian plume models taking advantage
of their simplicity and computational efficiency (Wang et al., 2020). An alternative but less explored option is to simulate the





plume with a full 3D atmospheric transport model. Such models can more realistically describe atmospheric transport and mixing than a Gaussian plume model and thereby better capture the structure of real plumes. They can also better represent complex flow conditions and temporal changes associated with the evolution of the atmospheric boundary layer. However,

accurately representing small-scale plumes is extremely challenging, because small errors in wind direction may create a simulated plume that does not overlap with the real plume. Furthermore, plumes are often turbulent, in which case even a perfect model will never be able to exactly match the observed plume due to the stochastic nature of turbulence. Traditional inverse emission estimation methods relying on a point-by-point comparison between simulated and observed $CO_2$ may therefore be inappropriate but more advanced non-local methods, as suggested by Farchi et al. (2016), may be required.

In May/June 2018, the $CO_2$ plumes of two large coal-fired power plants, Bełchatów in Poland and Jänschwalde in Germany, were observed with aircraft in-situ and remote sensing measurements in the context of the CoMet campaign (Fix et al., 2018; Gałkowski et al., 2021; Fiehn et al., 2020; Krautwurst et al., 2021; Wolff et al., 2021). These measurements provide a unique opportunity to study the capability of atmospheric transport models to simulate such plumes in a realistic manner and to define optimal sampling and modelling strategies for emission quantification.

Since several research groups were already performing or planning to perform simulations for these power plants, a coordinated effort was undertaken to compare the different models operated by the groups. A joint modeling protocol (Supplement S2) was created to harmonize the setup of the models (simulation periods, domains, location and intensity of the source) and the output (data format, variables, output grid) as much as possible in order to simplify the data analysis and to make the results comparable. Finally, six research groups operating six different atmospheric transport models agreed to perform simulations

following this protocol and to contribute to the present study. Our study includes five different Eulerian transport models but only one Lagrangian dispersion model. A similar model evaluation study including other Lagrangian models was recently presented by Karion et al. (2019). The present study complements their analysis by focusing specifically on emissions from power plants rather than on surface emissions.

The overall aims of this study are to

– Evaluate the model simulations against in-situ and remote sensing observations with respect to selected meteorological parameters and $CO_2$ concentrations.

– Analyse how the spatio-temporal variability and dispersion of the plumes are represented by the different models operating on a wide range of resolutions and provide recommendations for optimal model setup.

– Analyse how well emissions can be quantified from future CO2M satellite observations using two well-established

methods, the cross-sectional flux and integrated mass enhancement method.

– Provide recommendations for future measurement campaigns to optimally support the validation of model simulations and satellite observations.





## 2 Aircraft measurements of power plant plumes

In May/June 2018, the CoMet 1.0 (Carbon Dioxide and Methane Mission) intensive measurement campaign was conducted
to study $CH_4$ and $CO_2$ emissions from hot spots in Europe. A particular focus was placed on methane emissions from coal mining and other industrial activities in the Upper Silesian Coal Basin in Poland (Fiehn et al., 2020; Kostinek et al., 2021; Krautwurst et al., 2021). Three aircraft were operated during the campaign, two by the German aerospace center (DLR) and one by the Freie Universität Berlin (FUB). One of the goals of the campaign was to evaluate the lidar system CHARM-F (Amediek et al., 2017), an airborne demonstrator of the upcoming satellite mission MERLIN (MEthane Remote sensing lidar
missioN) (Ehret et al., 2017), and to investigate its capabilities to detect atmospheric gradients in vertical columns of $CO_2$ and $CH_4$ as well as plumes of individual sources. Another goal was to evaluate the synergistic use of airborne remote sensing and in-situ measurements for source detection and quantification.

One of the aircraft, the DLR-Cessna, was equipped with in-situ instruments and mostly flew in the atmospheric boundary layer (ABL). The two other aircraft, the DLR-HALO and the FUB-Cessna, primarily flew at constant altitude above the ABL to
measure vertical columns of $CH_4$ and $CO_2$ with active and passive remote sensing using the CHARM-F lidar and the MAMAP spectrometer (Gerilowski et al., 2011; Krautwurst et al., 2021), respectively. An overview of the airborne instruments and of the measurements used in this study is presented in Table 1.

To evaluate the capability of quantifying emissions of large $CO_2$ point sources with different measurement techniques and sampling strategies, the plumes of two of the largest coal-fired power plants in Europe were sampled on three different days
during the campaign. The Jänschwalde power plant in Germany was visited on 23 May 2018 with the two aircraft equipped with remote sensing instruments, the DLR-HALO and the FUB-Cessna. The FUB-Cessna aircraft also carried an in-situ instrument and performed multiple transects through the plume at different altitudes in the ABL. The Bełchatów power plant in Poland was visited twice, on 29 May 2018 with the DLR-HALO only and on 7 June 2018 with all three aircraft. The in-situ and remote sensing data collected on 7 June are the most comprehensive data set used in this study, while the comparatively small data set
collected on 29 May 2018 is not analyzed here.

The Bełchatów plant is the largest coal-fired power plant in Europe and one of the five largest in the world. In 2018, it released a total of 38.4 Mt $CO_2$/yr to the atmosphere according to the European Pollutant Release and Transfer Register (E-PRTR), approximately the same amount as the emissions of the country of Switzerland as officially reported to UNFCCC. $CO_2$ is released through two 299 m tall stacks. The power plant Jänschwalde is the third largest in Germany and the fourth largest in Europe with a total reported emission of 23.1 Mt $CO_2$/yr in 2018. Different from Bełchatów, its emissions are not
released through stacks but through six out of its nine cooling towers, because the modern flue gas treatment reduces the exhaust temperatures to a level that is too low to be vented through stacks (Busch et al., 2002). An overview of the two power plants, their total $CO_2$ emissions in 2018, and of the stack parameters used for plume rise calculations in this study is presented in Tab. 2. It should be noted that the coordinates are not identical to those reported in E-PRTR but were selected as the center of all emitting stacks or cooling towers. The difference between the reported address and the true location of the source was
about 800 m for Bełchatów and about 300 m for Jänschwalde.





## 2.1 Model systems

Simulations were conducted with six different state-of-the-art atmospheric transport and dispersion models with horizontal resolutions between 2 km and 200 m. Two of the models, COSMO-GHG and WRF-GHG, are mesoscale non-hydrostatic

numerical weather prediction (NWP) models extended with the capability to simulate the transport, emissions and atmosphere-biosphere exchange fluxes of greenhouse gases. Three of the models, WRF-LES, ICON-LEM and EULAG, are Large-Eddy-Simulation (LES) models, in which turbulent eddies larger than a certain filter width are explicitly resolved, whereas the smaller, less energetic scales are parameterised (Heus et al., 2010). The last model, ARTM, is a Lagrangian Particle Dispersion Model driven by prescribed vertical profiles of wind and turbulence depending on atmospheric stability. Although all models

are able to resolve the plumes in quite some detail (i.e. model grid spacings are small compared to the size of the plume), the different types of models and the wide range of resolutions varying over two orders of magnitude in terms of grid cell area, allow us to study the capabilities and limitations of different model concepts and to investigate the impact of resolution on the characteristics of the plumes. The LES models may be considered as a reference, as they have the most realistic representation of atmospheric turbulence. However, they are computationally expensive and their results critically depend on the specific

setup and forcing data. A summary of the participating models is presented in Tab. 3 and brief descriptions are provided in the following.

COSMO-GHG is based on the NWP and regional climate model COSMO (Baldauf et al., 2011), which was developed by a consortium of European weather services under the lead of the German weather service (DWD). The GHG extension allows for simulating the transport of passive trace gases and their emissions and surface exchange fluxes (Liu et al., 2017; Brunner

et al., 2019; Kuhlmann et al., 2019). An online emissions module was developed for a flexible treatment of anthropogenic emissions from point and area sources (Jähn et al., 2020) and was used here to prescribe vertical emission profiles for the two power plants. COSMO-GHG was run in a version optimized for execution on Graphical Processing Units (GPUs) (Fuhrer et al., 2014; Brunner et al., 2019). Meteorological initial and boundary conditions were taken from operational COSMO-7 analyses of the Swiss weather service MeteoSwiss at approximately 7 km horizontal and 1 hour temporal resolution. The domain of

COSMO-7 covers much of Europe and is nested into operational IFS analyses of the European Centre for Medium Range Weather Forecast (ECMWF). Within the model domain, the meteorology of COSMO-GHG was nudged toward observations from surface stations, radiosondes, and commercial aircraft using the scheme of Schraff (1998).

In this study, two distinctly different configurations of WRF were used, WRF-GHG and WRF-LES. The backbone of both is the Weather Research and Forecast model WRF (Skamarock et al., 2008) operated with the Advanced Research WRF (ARW)

core. WRF-ARW is a state-of-the-art Eulerian NWP model developed in a collaboration of several US research institutions led by the National Center for Atmospheric Research (NCAR), and integrates the non-hydrostatic, fully compressible Euler equations in flux form on a terrain-following mass-based vertical coordinate. The governing equations are expressed as perturbations from a hydrostatically balanced reference state. The WRF model can be applied from global scale to microscale, where atmospheric processes can be effectively downscaled through one- or two-way nesting. In both cases the system is operated





as a limited area model, using meteorological boundary conditions of a larger scale modelling system, namely the operational HRES IFS forecast from ECMWF, downloaded at 0.125°x 0.125°horizontal, L137 vertical resolution.

For the WRF-GHG configuration, the WRF-Chem add-on with the GHG option (Beck et al., 2011; Zhao et al., 2019) was used, allowing for the online simulation of the emission, transport and mixing of $CO_2$. All GHG tracers are treated as chemically inert (i.e. passive). WRF-GHG was run in a one-way nested setup with a parent domain spanning Europe at

10 km x 10 km horizontal resolution and a nested fine-grid domain at 2 km x 2 km resolution. We used the classic WRF pressure-based terrain-following vertical coordinate, with the model top at 5 kPa (approximately 21 km a.m.s.l.) and 85 vertical levels, with increased level density in the ABL. There were typically 33 levels below the altitude of 2 km. The internal time-step of the nested domain was set to 10 seconds. Details of the applied configuration are: WRF single-moment 5-class microphysics scheme, Rapid Radiative Transfer Model longwave radiation (RRTM) scheme, Dudhia shortwave radiation scheme and Grell-

Freitas cumulus parameterization (in the coarse domain only). Land surface was simulated using the Community Land Surface Model version 4. The Planetary Boundary Layer (PBL) was parameterized using the Mellor-Yamada Nakanishi and Niino 2.5 (MYNN) scheme, with the MM5 similarity surface layer parameterization. Similar to the setup of Ahmadov et al. (2007, 2009), the computations were performed as a series of 30 h simulations, with reinitialisation of meteorological fields every 24 hours (at 18:00 UTC) using the last available IFS forecast data, with subsequent recycling of the tracer fields at midnight (00:00

UTC), using the output from the end of the previous cycle.

In the WRF-LES configuration, WRF version 3.8.1 was operated with three nested domains with horizontal resolutions of 5 km, 1 km and 0.2 km, respectively. The outer domain used operational IFS analyses from ECMWF as meteorological boundary conditions with a horizontal resolution of 9 km and 6-hourly temporal resolution. The inner domain was run as a Large Eddy Simulation (LES) to resolve local turbulence. The implementation of passive $CO_2$ tracers in WRF-LES was

applied following the methodology of Blaylock et al. (2017) and used in Wolff et al. (2021) for simulations of the Jänschwalde coal-fired power plant.

The ICON (ICOsahedral Non-hydrostatic) model (Zängl et al., 2015) is a joint project of DWD, the Max Planck Institute for Meteorology (MPI-M) and their partners. For this study, ICON 2.4.0, coupled to the Modular Earth Submodel System (MESSy; Kern and Jöckel, 2016) was used. The spatial and temporal variation of passive tracer emission in the simulation

was controlled by the MESSy interface, whereas the transport of the tracers was handled by ICON. The simulations were performed in a limited area configuration with ICON running in LES mode (ICON-LEM; Dipankar et al., 2015). The large eddy simulations were driven by limited area ICON simulations over Germany and Poland, respectively, with a grid-spacing of approximately 2.5 km. Initial and boundary conditions for these simulations were provided from 6-hourly operational IFS analyses (ECMWF, 2020).

EULAG is NCAR's generic numerical framework for solving geophysical flow equations for a wide range of scales and applications. It allows solving the equations of fluid motion in either the Eulerian or the semi-Lagrangian mode (Prusa et al., 2008). The code has been used, in particular, to simulate turbulent flows in LES mode. EULAG is a research code that allows multiple adaptations based on particular user needs. The LES version used here solves the anelastic Navier-Stokes equations in the Eulerian form (Wyszogrodzki et al., 2012). Further model adaptations were performed for the needs of $CO_2$ modelling.



In particular, the model was coupled with output from the mesoscale model COSMO-GHG with several meteorological output fields from COSMO-GHG used to initialize the EULAG simulation as well as to force the model throughout the simulation. The COSMO-GHG fields provided on a 1 km ×1 km grid every 60 minutes (every 15 minutes during the period of CoMet flights) were interpolated to the EULAG spatial grid and time steps. The model domain included 400 x 300 grid points with a resolution of 0.003° (longitude) x 0.002° (latitude), which corresponds roughly to 208 x 220 meters. The domain was centered

on the power plant to allow the build-up of high resolution up-wind circulation in the model domain. The vertical resolution was 50 m. With 60 model levels, the model extended to 3000 m above the surface, well above the top of the ABL. The time step was 2 s, with model output stored every 15 minutes.

The Atmospheric Radionuclide Transport Model (ARTM) is an LPDM developed by the "Gesellschaft für Anlagen- und Reaktorsicherheit (GRS) gGmbH" in 2007 on behalf of the Federal Office for Radiation Protection (BfS) of Germany. It is

based on AUSTAL2000 (Janicke and Janicke, 2013), a widely used dispersion model for conventional tracers in Germany, and designed for modelling the dispersion of radionuclides emitted from nuclear facilities under routine operation on an annual time scale. Here, ARTM version 3.0.0 was used, which employs the same wind and turbulence models as version 2.8.0 (Hanfland et al., 2022) but with the ability of specifying the mixing layer depth as given in the modelling protocol. The spatial resolution of ARTM was limited by the maximum horizontal number of grid cells (300 x 300) and their maximum horizontal size

(666 m x 666 m). The temporal resolution was limited to 1 hour. ARTM runs a diagnostic wind field model creating wind and turbulence fields with homogeneous density for the simulation using meteorological data at a single location within the simulation domain. As such, the COSMO-GHG data were used for the Jänschwalde simulation. For the Bełchatów simulation, the COSMO-GHG data were only used before and after the measurement flight. During the flight, the data derived from the in-situ wind measurements were used to drive the model with two different wind directions in order to mimic the broad probability

distribution of the measured wind directions.

## 2.2 Modelling protocol

The protocol is provided in Supplement S2 and only the main points are summarized here. Each simulation needed to include a minimum set of three passive $CO_2$ tracers, CO2_PP_H, CO2_PP_M and CO2_PP_L, representing $CO_2$ emitted by the power plant (PP) according to three different scenarios in terms of vertical release height. In the low release scenario L, the emissions

were released at stack height without additional plume rise. In the reference scenario M, $CO_2$ was released according to a fixed vertical profile calculated using a plume rise model as described by Brunner et al. (2019). The plume rise model accounts for stack height and stack parameters such as flue gas temperature and volume flow (see Tab. 2 as well as for the specific meteorological conditions (wind speed, vertical stability) during the time of the aircraft flights. Meteorological conditions were taken from hourly COSMO-7 analyses of MeteoSwiss at the position of the respective power plant. The scenario H was

similar to M but corresponded to a release at a higher altitude computed as the maximum of all hourly plume rise calculations for the day of the flight and the previous day. The vertical profiles for the three scenarios (in meters above surface) are provided in Supplement S3. Each modelling group had to translate these profiles to the respective vertical coordinate system of the model.





Optionally, additional tracers could be simulated representing background $CO_2$, $CO_2$ emitted by all other anthropogenic

sources within the model domain, and $CO_2$ from biospheric uptake and release. Summing up these tracers with one of the

three power plant tracers should allow for a direct comparison with the in-situ $CO_2$ measurements. In this study, however, we

focus on the analysis of the power plant tracers only and compare the simulations with observed $CO_2$ plume enhancements

above a local background. The $XCO_2$ remote sensing data from MAMAP and CHARM-F were already provided as deviations

from a local background.

All simulations were required to cover at least the day of the flight and the previous day but were free to include additional

days of spin-up. Model output had to be reported on a prescribed latitude-longitude grid for both a large domain (approx.

200 km x 200 km) with about 1 km resolution and for a small domain (approx. 60 km x 60 km) with about 200 m resolution.

The small domain was selected to be sufficiently large to cover all aircraft transects. The large domain also captures parts

of the plume more than 30 km downwind of the source that may still be detectable by a future satellite such as CO2M

(Kuhlmann et al., 2021b). Models running at very high resolution were only able to cover the small domain. In contrast to the

horizontal direction, no grid was specified for the vertical direction but output was reported in the native vertical coordinate

system of each model. The output had to be produced in a standardized netCDF format and had to include both meteorological

variables (pressure, temperature, specific humidity, horizontal wind components, geopotential) and the different $CO_2$ tracers.

An overview of the two mandatory simulations and the corresponding SMALL and LARGE output grids is presented in Tab. 4.

A map of the domains and the ground tracks of the three aircraft is shown in Fig. 1.

### 2.3 Model performance assessment

The model simulations were compared with each other and with the in-situ and remote sensing observations. The comparison

between models allows for assessing the influence of different model types, configurations and resolutions. It also allows

investigating how differences in meteorology such as wind speeds and depth and stability of the ABL affect the model results.

The comparison with observations allows evaluating how well the main characteristics of the plumes are reproduced and how

well the simulated meteorology captures the true situation.

It is important to note that in the presence of atmospheric turbulence the comparability between models and observations

is fundamentally limited due to the stochastic and chaotic nature of turbulence (Lorenz, 1969). The observations only pro-

vide snapshots and each simulation only represents a single realization. Repeating a simulation with slightly perturbed initial

conditions would produce a different plume evolution, with different patterns of meandering, stretching and thinning that char-

acterize a turbulent plume. It is therefore more meaningful to compare statistical properties such as width and amplitude of the

plumes rather than comparing models and observations point by point. Other properties could also have been investigated, such

as probability density functions or spectra of concentration fluctuations, but these properties are more sensitive to measurement

uncertainties, which differed strongly between remote sensing and in-situ measurements.





To compare plumes between the models and in-situ and remote sensing observations, we divided the flights into individual plume transects (see Supplement Figures S1-S5) and fitted a Gaussian to the $CO_2$ data along each transect:

$$c_p(y) = \frac{A}{\sqrt{2\pi}\sigma} \exp\left(-\frac{(y-\mu)^2}{2\sigma^2}\right) \tag{1}$$

with scaling constant $A$, shift $\mu$ and standard deviation $\sigma$. Flight coordinates (latitude, longitude) were translated into a Cartesian coordinate system (units of meters) with its origin placed at the position of the power plant. The variable $y$ describes the

distance (m) flown from the starting point of each transect. The parameters $A$, $\mu$ and $\sigma$ were estimated using a non-linear least squares Levenberg-Marquardt minimization method starting from an initial guess. The initial value of $\mu$ was set to the center of the transect, $\sigma$ to 2000 m, and the area integral $A$ to 500 for in-situ measurements (in ppm) and 0.1 for column measurements (in $\mathrm{mol\,cm^{-2}}$). When no solution was found, a three times larger initial value of $\sigma$ was chosen. In this way, the method almost always converged to a solution, though sometimes with large uncertainties. Uncertainties of all three parameters were also

obtained from the fit procedure.

We estimate the true plume width from the Gaussian fit as $\sigma \cdot cf$, where the geometric correction factor $cf = \cos(\mathrm{atan}(y_c/x_c))$ accounts for the fact that transects were not perfectly perpendicular to the plume axis. Here, $y_c$ and $x_c$ denote the coordinates of the plume center in the coordinate system centered on the power plant. Finally, for each transect, the start and end coordinates and times, the total length of the transect, the fit parameters and their uncertainties, and the location and distance of the plume

center from the power plant were stored in a YAML file. Plume amplitude was computed as the maximum of the Gaussian curve at the location $y = \mu$. In order to make the amplitude comparable between in-situ and column observations, the columns were converted from $\mathrm{mol\,cm^{-2}}$ to ppm assuming that the plume extends uniformly over the full depth of the ABL, which was estimated from the observations to be 175 hPa (from the surface at 200 m to the top of ABL at 1800 m a.m.s.l) deep for Bełchatów and 160 hPa (60 m - 1520 m) for Jänschwalde.

## 3 Results and Discussion

### 3.1 Maps of total column $XCO_2$

In order to compare the representation of the Bełchatów plume between the different models, Figures 2 and 3 show the evolution of total column $XCO_2$ fields (CO2_PP_M tracer) on 6 June 2018 from the early morning to the early afternoon. In all models except ICON-LEM (Fig. 3d-i), the plume is transported into a northwesterly direction at all times. In the early morning at

05 UTC (approx. 06 local time, CET), the plume is compact and laminar in almost all models. A fanning out is visible in some models, which is a consequence of the advection of the plume into different directions due to vertical shear. With the rise of the sun in the morning (at 3:33 CET on 6 June), the ABL slowly starts to grow and eventually encompasses the plume release height. At this point in time, the plume starts becoming turbulent.

The onset of turbulence is clearly visible at 09 UTC in the LES models (Fig. 3b,e,h) and the high-resolution WRF-GHG

simulation (Fig. 2h), whereas turbulence is still moderate in COSMO-GHG and the low-resolution WRF-GHG simulation (Fig. 2b,e). The plume reaches a highly turbulent state by 13 UTC around the time of the aircraft flights. The widest plumes





at this time of the day are simulated by COSMO-GHG, WRF-LES and EULAG (Fig. 2c,3c,l). In COSMO-GHG this is due to mixing of a small portion of the plume into the free troposphere, where wind direction was nearly opposite to the ABL. The same effect, though less pronounced, is also seen in WRF-LES.

The size spectrum of turbulence is wide enough that even NWP models like COSMO-GHG or WRF-GHG running at 1–2 km resolution are able to resolve the largest eddies. However, the variability in $XCO_2$ clearly grows with resolution, which is especially evident when comparing the two WRF-GHG simulations at 2 km and 0.4 km resolution, respectively (Fig. 2f,i). Another impact of resolution is that the plume expands much more quickly in the initial phase upon release in a low-resolution simulation, which is again best seen by comparing the two WRF-GHG runs. No turbulent structures are visible in ARTM.

Instead, the plumes mostly have a Gaussian shape, except for a fanning out at 05 UTC due to vertical wind shear. ARTM is forced with constant vertical wind profiles every 60 minutes. As a result, the plume can slightly change direction with distance from the source. In ARTM, the plume is only slightly wider at 13 UTC than at 09 UTC, but much wider as compared to nighttime (not shown). This shows that also ARTM accounts for increased turbulent dispersion during daytime, though the plume is significantly more compact than in the other models. Tests with different turbulence parameterizations indicate that

the standard configuration of ARTM tends to produce too narrow plumes (Hanfland et al., manuscript in preparation). Except for resolution, there seems to be no clear difference between NWP and LES models. The plume simulated by WRF-GHG at high resolution, for example, is structurally very similar to the plumes simulated by WRF-LES and ICON-LEM at comparable resolution.

    Similar maps for the plume of the Jänschwalde power plant on 23 May 2018 are presented in Supplement S1 Fig. S7. Only

results for 10:00 UTC are shown, which roughly corresponds to the time of the aircraft overpasses. At this time of the day, the turbulent structures of the plume were not yet as wide (only visible in the LES simulations) and the plume itself was less dispersed as the plume observed around noon at Bełchatów.

### 3.2   Qualitative comparison with in-situ observations

The DLR-Cessna flew a total of 12 transects through the Bełchatów plume at multiple levels and at three distances from the

source (Fig. S1 in the Supplement). The $CO_2$ measurements along these transects provide detailed insights into the horizontal and vertical extent of the plume. To compare the simulations with the observations, meteorological quantities and $CO_2$ mole fractions were interpolated to the flight track. In a first step, the 4-D model fields were interpolated in time and latitude-longitude space to produce vertical curtains along the flight track. In a second step, these curtains were interpolated vertically to the flight altitude to produce time series corresponding to the observations.

Curtains of $CO_2$ along the flight track are presented in Fig. 4 for all model simulations. The corresponding in-situ measurements are overlaid as colored circles with the same color scale. A constant background of 399.8 ppm was subtracted from the observations, which is 1 ppm higher than the lowest observed mole fraction. The first transect was flown at an altitude of 1000 m close to the source at a distance of about 9 km, followed by seven transects at 14 km distance starting at an altitude of 800 m and rising step by step to 1900 m. Another four transects were then flown at 26 km distance between 800 m and 1450 m

above sea level. Finally, the aircraft rose to an altitude of 2200 m well above the ABL.





The plume enhancements are clearly visible in the observations typically near the center of the horizontal transects. Elevated $CO_2$ mole fractions were also measured at the highest altitudes above about 1800 m, at the beginning of the flight around 13:05 UTC, later around 14:15 UTC, and especially after 14:50 UTC when the aircraft rose to 2200 m. These enhancements were due to higher background $CO_2$ above the ABL, which is typical for summertime when biospheric uptake by photosyn-

thesis reduces $CO_2$ in the continental ABL (Sweeney et al., 2015). These elevated values are therefore not reproduced by the simulated power plant tracers. An exception is the situation around 14:15 UTC, where likely a mixed signal of elevated $CO_2$ from background air above the ABL and from the plume was measured. Peak values of both $CO_2$ and other species like $CH_4$ were somewhat higher than observed elsewhere at similar altitudes in the free troposphere.

The multiple plume crossings are also visible in the simulated curtains (Fig. 4a-h). In all simulations, the plume essentially

extends from the surface to the top of the ABL, which suggests rapid vertical mixing in an unstable, convective ABL. Since the closest transect was at 9 km and typical wind speeds in the ABL were around 5 m/s (18 km/h), the simulated time scale of vertical mixing in the ABL was only of the order of 30 minutes. As a consequence, there is no clear difference between the tracers CO2_PP_H, CO2_PP_M and CO2_PP_L released at different altitudes. Fig. 4 shows the results for the reference tracer CO2_PP_M. Results for the other two tracers are presented in the Supplement (Fig. S8,S9).

The shape, width and vertical extent of the plume varies quite substantially between the models. The plumes are more strongly dispersed horizontally and less well confined at the top in COSMO-GHG and WRF-LES (Fig. 4a,c) as compared to WRF-GHG and ICON-LEM (Fig. 4b,d,e,f). In the latter two models, the plumes are sharply capped at the top of the ABL suggesting little exchange with the free troposphere aloft. In the high-resolution version of WRF-GHG, the ABL is about 100 m deeper than in the low-resolution version. The plume extends too high to about 2400 m in ARTM (Fig. 4g) because of

the coarse vertical resolution of the model output grid in the upper part of the domain. The top layer in ARTM extends from 2100 to 2400 m. The assumed mixing layer top of 2000 m above surface for the period of the aircraft flight allowed the plume to mix into the top layer and in this way to reach 2400 m. A finer vertical resolution in the upper part of the domain would likely have prevented this. In contrast to ARTM, the plume stays comparatively low in EULAG, mostly below 1500 m (Fig. 4h). This can be compared to the COSMO-GHG model, which provides the lateral forcing data (and surface sensible heat fluxes) for

EULAG. Both models show the main part of the plume at rather low altitudes, but the plume is even lower in EULAG and also more sharply capped at the top of the ABL compared to COSMO-GHG. EULAG simulated a more compact plume than COSMO-GHG in the horizontal direction as well.

Many of the differences between the models can be explained by differences in the structure of the ABL. Figure 5 presents curtains of potential temperature for all models with the observations overlaid. Consistent with the $CO_2$ curtains, the capping

inversion at the top of the ABL is much sharper in WRF-GHG and ICON-LEM (Figure 5b,d,e,f) than in COSMO-GHG, WRF-LES and EULAG (Figure 5a,c,g). In the latter models, the inversion is not only weaker but also more fuzzy, suggesting significant entrainment/detrainment at the inferface between the ABL and the free troposphere. This likely explains why parts of the $CO_2$ plume are advected in reverse direction, especially in COSMO-GHG (see Fig.2c). Compared to the observations, the top of the ABL is too high and too weakly stratified in WRF-LES (Fig. 5c), instead vertical stability starts increasing already

at about 1500 m so that only a small fraction of the plume mixes into the top of the ABL at 1900–2000 m. A similar conclusion





can be drawn for COSMO-GHG and especially EULAG, where stability starts increasing already well below 1500 m. In contrast, WRF-GHG and especially ICON-LEM show an almost perfectly neutral ABL up to the capping inversion. Despite a comparatively low ABL, the core of the plume extends higher up in these models. No curtain is shown for ARTM since turbulent mixing in this model is not constrained by a temperature profile but is prescribed depending on stability class. The
measurements indicate a top of the ABL at about 1900 m capped by a sharp inversion. WRF-GHG is the model that captures the ABL structure most accurately. Similar curtains of wind speed are presented in the Supplement (Fig. S10).

Plume transects at multiple vertical levels in the ABL were also performed by the FUB-Cessna aircraft at Jänschwalde during the second part of its flight on 23 May 2018. In the first part, the aircraft had flown above the ABL (close to its top) to sample vertical column transects of the plume with MAMAP. Curtains of $CO_2$ along the second part of the flight are compared with
the in-situ measurements in Fig. 6. No simulations are available for this flight from the high-resolution version of WRF-GHG and from EULAG. Compared to the observations, the plume is too wide and amplitudes are too low in COSMO-GHG and WRF-GHG (Fig. 6a,b), the two models with comparatively low resolution. These models also underestimate the vertical extent of the plume, which was clearly detectable in the observations up to about 1500 m, whereas the simulated plumes only extend to about 1300-1400 m. Somewhat surprisingly, the observed plume was stronger during the first three transects at the highest
flight levels in the ABL, which is opposite to the strengths of the plumes simulated by COSMO-GHG and WRF-GHG. This behavior is quite well reproduced by WRF-LES, which however simulated a plume with a more complex structure compared to the observations suggesting that it might have overestimated turbulence intensity. A similarly complex structure with two or more sub-plumes was also simulated by ICON-LEM. As for Bełchatów, the plume is displaced in the ICON-LEM model suggesting that winds were not accurately captured. This is supported by curtains of wind speed (see Supplement Fig. S12),
which show a strongly different behavior of ICON-LEM as compared to other models. Note that no temperature and wind measurements are available for this flight.

Time series of observed and simulated $CO_2$ along the DLR-Cessna flight at Bełchatów are presented in Fig. 7. Both observations and simulations were averaged over 5-s intervals along the flight track, which corresponds to a distance of about 350 m. The observations reveal sharp peaks of more than 40 ppm in the first transect and gradually wider and lower peaks down to
about 10-15 ppm in the last four transects. The width and amplitude of the simulated peaks varies considerably. COSMO-GHG consistently underestimates the peaks (Fig. 7a), especially at the higher flight levels, due to insufficient mixing into the upper ABL as mentioned before. WRF-GHG underestimates the plume amplitude in the low resolution setup (Fig. 7b), but captures and partly overestimates the amplitudes in the high resolution setup (Fig. 7d). The plume transects are quite well represented in WRF-LES (Fig. 7c), except for the first transect, where the peak amplitude is strongly underestimated. A similar underes-
timation for this transect is also present in all other models, which may indicate a turbulent structure of unusually high $CO_2$ concentrations encountered during the flight. In ICON-LEM, the plumes tend to be narrower than observed (Fig. 7e,f), and they are displaced due to the erroneous wind direction. ARTM reproduces plume location and amplitude quite accurately, but the plumes tend to be narrower than observed despite the usage of two alternating wind directions in the simulations, which generated additional plume spread. Finally, EULAG quite well captures the plume at the lowest flight level but fails to reproduce
the observed peaks at the higher levels due to insufficient vertical extent of the plume as mentioned before.





Corresponding time series of potential temperature and wind speed are presented in the Supplement (Figs. S15 and S16). While average wind speeds are quite accurately captured by COSMO-GHG and WRF-GHG, they are slightly overestimated by WRF-LES and strongly overestimated by ICON-LEM. In EULAG, mean wind speeds are close to observations, but fluctuations are too large suggesting that turbulence was too strong in this model.

Time series of in-situ observed and simulated $CO_2$ along the FUB-Cessna flight at Jänschwalde are presented in Fig. 8. The plumes are quite well represented by WRF-LES, they are generally too wide and underestimated by WRF-GHG, they are underestimated by COSMO-GHG and ARTM during the first three but overestimated during the last two transects, and they are misplaced by ICON-LEM. Similarly as for Bełchatów, there is a large variability between model results suggesting that it is very difficult to represent the observed plumes in all details.

### 3.3  Qualitative comparison with vertical column $XCO_2$ observations

Models may be more successful in reproducing vertical columns as these are much less sensitive to vertical transport and mixing in the ABL. Time series of vertically integrated $CO_2$ [$\mu$mol cm$^{-2}$] from the different models interpolated in time and space to the flight tracks of the FUB-Cessna at Bełchatów (7 June 2018) and Jänschwalde (23 May 2018) are compared in Figures 9 and 10 with corresponding vertical columns measured by MAMAP.

In contrast to in-situ $CO_2$, COSMO-GHG reproduces the observed total columns at Bełchatów quite accurately (Fig. 9a). ICON-LEM, in contrast, tends to underestimate the total columns and partly misses the plumes due to a wrong wind direction (Fig. 9e,f). The underestimation can be explained by a strong overestimation of wind speeds in the ABL (see Supplement Fig. S16). Furthermore, the plumes are much narrower than observed, which was already noticed in the comparison with the in-situ measurements and could also be a consequence of too high wind speeds. Too narrow plumes are also simulated

by ARTM during the first half of the flight (first 7 transects in Fig. 9g between 12:20 and 13:27 UTC), but different from ICON-LEM, this leads to an overestimation of peak amplitudes. During the second half (last 7 transects), the plumes simulated by ARTM agree much better with the observations. WRF-LES and WRF-GHG capture the plume transects quite well and mostly at the correct position (Fig. 9b,c,d). Peak amplitudes are better matching the observations in the high-resolution version of WRF-GHG. EULAG reproduces the total columns much better than the in-situ $CO_2$, because the underestimation of the

vertical extent of the plume does not affect the columns. The plume widths and amplitudes are very well matched except for an overestimation of the amplitude and underestimation of the width during the first two transects closest to the source. For Jänschwalde (Fig. 10), the overall quality of the agreement with the observations is similar but the results for the individual models are somewhat different. WRF-LES and especially ICON-LEM misplaced the plume and therefore missed it on selected transects.

A comparison with the columns measured by the CHARM-F Lidar at a distance of only about 3-4 km downwind of the Bełchatów power plant is presented in Fig. 11. To convert differential absorption optical depths (DAOD) as measured by CHARM-F into vertical columns, a differential absorption cross-section of $7.27 \times 10^{-23}$ $cm^2$ was assumed (Wolff et al., 2021). The observations are rather noisy but the enhancements during the four plume transects are clearly visible. Except for ARTM





and EULAG, the models tend to underestimate the plume amplitudes quite substantially, mainly due to too wide plumes. The
corresponding figure for Jänschwalde is shown in Supplement Fig. S17.

## 3.4   Evaluation of statistical properties of the plume

In order to compare characteristic properties of the plumes between simulations and observations, a Gaussian curve was fitted
to each aircraft transect as described in Sect. 2.3. Although most of the plume transects did not reveal a classic bell shape, it
was often possible to determine the fit parameters of the Gaussian with reasonably low uncertainty. Examples are presented in
Supplement Fig. S6.

A summary of the observed and simulated plume characteristics is presented in Fig. 12 as a function of distance from the
Bełchatów power plant. Width and amplitude of the plume were determined for both in-situ $CO_2$ along DLR-Cessna transects
and for column $CO_2$ enhancements along the FUB-Cessna and HALO transects. The corresponding measurements are shown
as open circles, squares and diamonds, respectively. The model results are presented as filled colored symbols. Although the
same transects were considered, the distance from the source varies between observations and models, because for each plume
the geometric distance between (fitted) plume center and power plant was determined. As described in Sect. 2.3, plume widths
were geometrically corrected to represent the width perpendicular to the plume axis and vertical columns were converted to
mole fractions to enable a joint analysis with the in-situ measurements.

For both, observations and models, the plume width generally increases and the amplitude decreases with distance, as
expected. However, between about 13 km and 26 km, there is no clear tendency in plume width, neither in the observations nor
in the model simulations. A possible reason could be that the plume was not fully covered by the transects at 26 km. This is
true for some of the simulated plumes due to the limited model domain, but it is not obvious for the observations. However,
the fact that plume amplitude changed only little suggests that the plume did indeed not grow between 13 km and 26 km.
Overlaid in the figure are plume width estimates from a classical Gaussian plume model following Briggs (1973). The two
lines describe an average behaviour of turbulent plumes under very unstable (stability class A) and slightly unstable (stability
class C) atmospheric conditions. The observed plume growth up to a distance of 15 km is quite consistent with the Gaussian
plume model for very unstable conditions (grey dashed line), but at 26 km distance the observed plume is almost twice as
narrow than expected.

The model results show a wide range in both width and amplitude but the mean model behavior is quite consistent with the
observations. In the near field up to distances of about 8 km, models with lower resolutions (COSMO-GHG and WRF-GHG)
tend to show wider plumes than models with higher resolutions (ICON-LEM, WRF-GHG-HR, WRF-LES, EULAG). The
Lagrangian model ARTM, which can represent the source as a true point release without averaging over the extent of a grid
cell, simulated a very compact plume in the near-field that is clearly narrower than the plume observed by both MAMAP and
HALO. Also the Eulerian models with very high resolutions simulated a too narrow plume in the near field. A possible reason
is that the plumes were released at a single point above the power plant, whereas in reality the release occurred from two stacks
separated by 350 m. Furthermore, the plume had likely spread horizontally already during plume-rise, a process that was not
considered in the simulations where $CO_2$ was released from a single horizontal location (or grid cell) above the chimney.





The observations in the near-field, which primarily originate from MAMAP and HALO, show a rapid growth of the plume up to a width of about 2 km at a distance of 5 km, suggesting a strongly turbulent nature of the plume. In fact, the second and third closest transects from MAMAP show a split of the plume into two and three parts, respectively. Also the closest transect observed from the DLR-Cessna at 9 km shows a double-structured plume. The observed plume was strongly displaced to the north away from the main plume axis, suggesting that a turbulent eddy had pushed it northwards upon release from the power plant. Since this was not reproduced by any of the models, the model symbols corresponding to this transect appear in the figure at a much shorter distance around 6-7 km.

The evolution of plume amplitudes shows a somewhat more robust behavior than plume widths, with a clearly decreasing trend up to 13 km, but only a small further decrease up to 26 km. A possible explanation for the higher robustness could be that the fitting of plume amplitude is less sensitive to incomplete coverage of the plume within a transect. Again, the models with resolutions of 1 km or coarser show a much faster dilution in the near-field and a corresponding underestimation of plume amplitude. This is especially evident when comparing the results of WRF-GHG, which was run at 2 km and 400 m resolution. The high-resolution version is much more consistent with the observations. The high-resolution models ARTM and EULAG tend to overestimate the amplitudes in the near-field, consistent with their underestimation of plume width in this range.

At larger distances, the plume amplitudes are largely consistent between the models and the observations. However, COSMO-GHG consistently underestimates plume amplitudes, suggesting a too rapid dispersion not only near the source but also at larger distances. Despite the fact that the plumes simulated by ICON-LEM are too narrow, their amplitudes are quite comparable to the observations, which is likely due to the too high wind speeds of this model as mentioned earlier.

The same analysis was also performed for the measurements collected during the FUB-Cessna flight at Jänschwalde on 23 May 2018 (Fig. 13). No results from CHARM-F on HALO are included here as it was difficult to fit a Gaussian to these observations. To support the visual comparison with the results at Bełchatów, the same axis ranges were used. In comparison to Bełchatów, the plume at Jänschwalde remained more compact in both, the observations and the simulations, which is likely due to a combination of lower turbulence and higher wind speeds. The evolution of plume width is quite consistent with a Gaussian plume model for weakly unstable conditions (grey dotted line). Even more obvious as for Bełchatów, the two comparatively coarse models WRF-GHG and COSMO-GHG overestimate plume width in the near-field, but agree better at distances larger than 15 km from the source. For the in-situ transects (between 10 and 12 km), WRF-GHG overestimates the plume widths and underestimates the amplitudes quite substantially, whereas the agreement for the vertical column transects is much better. ICON-LEM tends to underestimate the plume width, though no comparison could be performed at distances larger than 10 km because the simulated plume moved out of the measurement transects rather quickly due to the wrong wind direction. WRF-LES performed the best matching both, the observed plume widths and amplitudes quite accurately.

## 3.5 Emission quantification with a CO2M like satellite

In this section, we generate synthetic total column $CO_2$ observations from the model outputs mimicking those of a future CO2M satellite and analyse two popular emission quantification methods applied to these synthetic satellite images. The main purpose is to determine how well the true emissions can be estimated from single CO2M satellite overpasses assuming that





the models provide a realistic representation of such plumes. We also analyze how diurnal variability in ABL structure and measurement noise affects the ability to quantify emissions.

The observations are generated by reducing the resolution of the output to 2 km x 2 km (through averaging over multiple
output grid cells) and adding Gaussian random noise corresponding to a low (0.5 ppm) and a high noise (1.0 ppm) instrument scenario of CO2M (Sierk et al., 2021). Assuming a depth of the atmosphere of 950 hPa, this corresponds to a noise of $1.67 \times 10^{-5}$ mol cm$^{-2}$ and $3.34 \times 10^{-5}$ mol cm$^{-2}$ in total column $CO_2$, respectively. The two quantification approaches are the cross-sectional flux and the integrated mass enhancement (IME) method, which were identified by Varon et al. (2018) as comparatively robust methods.

The two methods are illustrated in Fig.14 for the example of a plume at Bełchatów on 7 June 2018 12 UTC simulated by the WRF-GHG model in the high-resolution (HR) configuration. In the low-noise scenario (Fig.14a), the plume signal clearly stands out from the background noise. In the high-noise scenario (Fig.14b), in contrast, the noise partly obscures the plume signal. Since the simulated plume amplitude linearly scales with the emission strength, the high-noise scenario would be identical to a low-noise scenario for a two times smaller emission source.

The cross-sectional flux method integrates total column $CO_2$ (kg m$^{-2}$) along a cross-section approximately perpendicular to the plume axis and obtains the emission as the product of this line density (kg m$^{-1}$) with an effective wind speed perpendicular to the cross-section (m s$^{-1}$). For simplicity, we chose exact north-south cross-sections together with the east-west wind component $U$. In order to obtain a representative wind speed, the wind component $U$ was evaluated in the center of the plume transect (filled black circles Fig. 14) and averaged over the pressure range 925-875 hPa (approx. 800-1200 m above sea level),
which approximately corresponds to the center of the daytime ABL. Similar to Kuhlmann et al. (2021b), we computed the emission as the average over multiple cross-sections (dashed lines) in order to make better use of the imaging capability of a future CO2M satellite. Only cross-sections for which the fitted Gaussian curve was fully ($\pm 2\sigma$) inside the model output domain were included in the average. In the example, all cross-sections fulfilled this criterion.

In case of IME, the integrated mass enhancement (i.e. the total mass of $CO_2$ within the plume) was determined from all pixels
above a given threshold (white crosses in Fig. 14). As recommended by Varon et al. (2018), the image was first smoothed with a Gaussian filter of width 200 m ($1\ \sigma$) in order to limit erroneous detection of pixels outside the plume due to measurement noise. The filtering substantially stabilized the detection of the plume, especially for the high-noise case (Fig. 14b). The emission $Q$ was then computed as (Varon et al., 2018):

$$Q = \frac{U_{eff}}{L}\text{IME} \tag{2}$$

where $U_{eff}$ is the effective wind speed and $L$ a characteristic length scale of the plume. The ratio $L/U_{eff}$ represents the residence time of $CO_2$ within the detected plume. A possible measure of the length scale $L$ is the square root of the area of the detected pixels (Varon et al., 2018). It is important to note that the exact choice of threshold and length scale affects the effective wind speed. In contrast to the cross-sectional flux method, the effective wind speed may be a non-linear function of the true transport speed of the plume and first needs to be calibrated to obtain an unbiased estimate of $Q$. Varon et al. (2018)
suggested to perform LES model simulations to determine this relationship. Here, we took the wind speed (square root of sum





of squared $U$ and $V$) at the position of the power plant averaged over the same altitude range as for cross-sectional flux method. In order to bring the estimated emissions in close agreement with the truth, this wind speed had to be multiplied by a factor 0.75. This potential caveat of the method will be discussed later.

Fig. 15 presents a comparison of the results of the two methods. To enable a fair comparison, the image was first smoothed
before applying the cross-sectional flux method in the same way as for the IME method. The figure shows the emissions estimated from the simulated plumes at Bełchatów for all 24 hours of 7 June 2018. For both methods, the scatter between the models is lower around noon-time than at night, which is a result of the strong vertical mixing during daytime. Approximating the effective transport speed by a wind speed in the middle of the ABL seems to be a good approach under these conditions. At night, conversely, the results are more sensitive to the altitude range over which the wind speeds are averaged because of
vertical wind sheer and a more limited vertical extent of the plume.

A summary of the the performance of the two methods for midday (9-15 UTC) averaged fluxes is presented in Tab. 5. Overall, the results of the two methods are comparable, with the cross-sectional flux method producing slightly more robust results (smaller scatter between the model results). For both methods, the multi-model mean bias is mostly well below 10 percent and the standard deviation is of the order of 20%, slightly higher for the high-noise scenario than for the low-noise
scenario. One reason for the fluctuations between the model results is measurement noise. However, even without any noise the standard deviation is still about 17% of the true value (see Tab. 5). A second reason is that the assumed 925-875 hPa average wind speed is only an approximation of the true transport speed. Finally, the fluxes through vertical cross-sections are not constant in time and space due to turbulent fluctuations. Averaging over multiple cross-sections reduces this variability but does not eliminate it. For the low noise scenario, the standard deviation of the emissions estimated for the 10 individual
transects is of the order of 20% to 30% depending on the model. Averaging over 10 transects reduces this uncertainty by roughly a factor of 3 ($\sqrt{10}$). For a satellite like OCO-2 with a narrow swath of only 8 km, the possibilities for averaging are much more limited, such that substantial uncertainties of the order of 10%-20% due to turbulent fluctuations alone have to be expected. The same applies to the planned lidar satellite MERLIN, which will measure along a very narrow ground-track. Wolff et al. (2021) therefore concluded that emissions can be better quantified from MERLIN under less turbulent conditions at
night and in the early morning than at midday. However, our results suggest that this is only true if the height of the plume and the corresponding wind speed are well known. These parameters are likely more difficult to estimate for a vertically structured atmosphere at night than for a well-mixed ABL during daytime.

As mentioned earlier, the wind speed had to be scaled for the IME method by a factor 0.75 to obtain emissions close to the truth. The estimation of this scaling factor comes at the price of an additional uncertainty that is not present in the cross-
sectional flux method. In practice, the relationship between true and effective wind speeds may be determined from multiple observations over a known source or from realistic simulations with a high-resolution transport model. However, this (non-linear) relationship likely depends not only on wind speed but also on the turbulent state of the atmosphere, which makes the calibration a challenging multi-dimensional problem.





# 4   Conclusions

Six atmospheric transport models differing in type and resolution were used for simulating the $CO_2$ exhaust plumes of two large coal-fired power plants, Bełchatów in Poland and Jänschwalde in Germany, following a common protocol. The simulations were compared among each other and evaluated against a comprehensive data set of airborne in-situ and remote sensing observations collected on two fair-weather days in May and June 2018 by the CoMet measurement campaign.

The simulations indicate that, with the growth of the ABL in the morning, the plumes evolved from compact laminar plumes
at night into much wider, highly turbulent plumes during the day. The turbulent nature of the daytime plumes was not only captured by the high-resolution (200-600 m) LES models but also by the mesoscale NWP models operating at 400 m - 2 km horizontal resolution, though turbulent structures were increasingly smoothed out and not well represented anymore at 2 km resolution.

Characteristic properties of the plumes such as vertical extent and horizontal dispersion differed substantially between the
models. Consistent with the observations, the simulated plumes extended over almost the whole depth of the ABL during daytime. As a consequence, the exact altitude of the release of $CO_2$ in the models did not have a strong impact on the results in the early afternoon when most of the measurements were collected. Nevertheless, differences in the vertical stability in the upper parts of the ABL and the strength of the capping inversion had a significant effect on the simulations near the top of the ABL. WRF-GHG and ICON-LEM simulated an almost neutral ABL with a sharp inversion for the Bełchatów case in good
agreement with the observations, whereas WRF-LES, COSMO-GHG and EULAG showed a fuzzier and wider ABL top and an increase in stability already well below the capping inversion. This dampened the vertical expansion of the plume and led to an underestimation of plume height in COSMO-GHG and especially EULAG. Vertical plume extent was also underestimated by some models at Jänschwalde, including WRF-GHG, which performed much better at Bełchatów. Differences in vertical dispersion between different models was also found in the study of Karion et al. (2019) to be a major driver of differences in
simulated concentrations and emission sensitivity. Similarly, Katharopoulos et al. (2022) identified the turbulence description of the LPDM FLEXPART-COSMO as a main source of error when operating at high-resolution using the Bełchatów plume as a benchmark.

Simulations at resolutions coarser than about 1 km showed a too rapid dispersion of the plumes in the near field up to about 8 km downwind of the source, but the further dispersion was not systematically different from higher resolution models. The
high resolution LES models WRF-LES and ICON-LEM and the Lagrangian model ARTM, in contrast, simulated a too narrow plume in the near field, possibly because the plumes were released in the simulations from a single point rather than from multiple stacks and horizontal mixing (and displacement) during plume rise was not accounted for.

Overall, the COSMO-GHG model overestimated the dispersion. ARTM in contrast, simulated a generally too compact plume for the Bełchatów case when the ABL was very unstable but performed better for Jänschwalde. The plumes were also generally
too narrow in ICON-LEM, especially at Bełchatów, probably due to a significant overestimation of wind speeds for this case. Plume width was generally well represented at all distances by WRF-LES, but wind speeds were slightly overestimated at





Bełchatów. WRF-GHG showed too wide plumes when run at coarse resolution (2 km x2 km) but agreed much better with the observations when run at high resolution (400 m x 400 m).

The agreement with total column $CO_2$ measurements was usually better than with in-situ measurements, because errors in
the vertical distribution have only a minor impact on total columns. EULAG, for example, which showed a poor agreement with in-situ measurements at Bełchatów due to the underestimation of plume height, showed very good agreement with total columns from the MAMAP spectrometer and the CHARM-F lidar.

Based on the limited sample of only two measurement days, it is difficult to draw general conclusions on model performance or to even rank the models. Several simulations, for example the high-resolution version of WRF-GHG for Bełchatów and
WRF-LES for Jänschwalde, showed a remarkably high consistency with the observations, suggesting that power plant plumes can be simulated by both LES and NWP models in a very realistic way. However, the stochastic nature of turbulence puts fundamental limits to any point-by-point comparison. Good or bad agreement in a point-by-point comparison can be a matter of luck. In the ESA-funded project SMARTCARB2, an ensemble of 18 COSMO-GHG simulations with slightly different settings were performed for the Bełchatów case each producing a different realization of the turbulent plume. The results
revealed a large spread in model performance with correlation coefficients ranging from 0 to 0.8 depending on whether the simulated plume was structurally similar to the observed plume or not (Kuhlmann et al., 2021a).

Nevertheless, a few general conclusions can be drawn. Models with resolutions of 1 km or coarser tend to simulate wider plumes and significantly overestimate plume width in the near field at distances up to about 8 km from the source. Realistically representing turbulent structures of the plumes requires simulations at 1 km resolution or better. Model resolution appears to
have a larger impact on the results than differences in the treatment of turbulence between LES and NWP models. When run at comparable resolution, LES and NWP models showed very similar performance. The agreement of a model with observations critically depends on the set up and forcing of the simulation. Initial and boundary conditions from a meteorological analysis improves the representation of the meteorological situation. Additional assimilation of meteorological observations within the model domain (as in COSMO-GHG) or frequent re-initialization of the simulations from analyzed meteorological fields
(as in WRF-GHG and WRF-LES) can further improve the performance. ICON-LEM did not capture the weather situation well probably because it was forced by a free-running regional ICON simulation that was not sufficiently constrained by meteorological observations. The ARTM simulation at Bełchatów, in contrast, accurately captured the position of the plume as it was forced by observed winds from the aircraft.

The model simulations were used to generate synthetic CO2M satellite observations in order to analyze the capability
of CO2M to quantify emissions using two popular emission estimation methods, the cross-sectional flux method and the Integrated Mass Enhancement (IME) method. Assuming that winds in the middle of the ABL are a good approximation of the true transport speed of $CO_2$ in the plume, the emissions from Bełchatów can be estimated from a single overpass of CO2M with an uncertainty of about 20% with a bias of no more than a few percent. Because the satellite image was first smoothed (with a Gaussian filter) before applying the flux estimation, the uncertainty was only slightly higher for a CO2M
instrument scenario with high measurement noise (1.0 ppm) than for a low noise scenario. The performance of the IME and cross-sectional flux methods were very similar, but the IME method suffers from an additional uncertainty that is introduced

by the fact that wind speeds cannot be used directly but have to be translated into an effective wind speed. Averaging over multiple transects substantially improves the estimates in case of the cross-sectional flux method, because fluxes through individual 2 km wide transects fluctuate by 20%-30% due to turbulence. Such averaging will be possible for the upcoming

CO2M satellite constellation owing to its wide swath. Because turbulence is much reduced at nighttime, it seems attractive for an active lidar instrument like CHARM-F or the future MERLIN satellite to quantify emissions from measurements at night (Wolff et al., 2021). However, because the plumes are much more confined in the vertical at night, the results will critically depend on an accurate estimation of plume height in-situations where vertical wind shear is strong.

The combination of in-situ (chemical tracers and meteorological parameters) and remote sensing observations at varying

distances from the source collected during CoMet provided an excellent data set for evaluating the vertical and horizontal structure of the plumes as simulated by the models. Nevertheless, for future campaigns it would be desirable to sample power plant plumes under different meteorological conditions with stronger and weaker winds and turbulence and at different times of the day, including measurements at night. Furthermore, imaging spectrometry as planned for a forthcoming CoMet campaign could reveal much more details of the horizontal structure of the turbulent plumes and multiple overpasses could provide

critical insights into their dynamic nature.

*Data availability.* The MAMAP CH4 column anomalies are available from the authors upon request. The airborne in-situ measurements acquired by the DLR Cessna, the FUB Cessna and the DLR HALO and the CHARM-F measurements can be obtained from the authors or downloaded from the HALO database (https://doi.org/10.17616/R39Q0T, Deutsches Zentrum für Luft- und Raumfahrt, 2021).

*Author contributions.* DB wrote the manuscript, contributed to the protocol, and conducted the analyses. SH and GK performed, processed

and analyzed the COSMO-GHG simulations and together with EK developed the comparison and emission estimation concepts in the SMARTCARB2 and CoCO2 projects. BK and PJ performed and analyzed the ICON-LEM simulations, SW and CK the WRF-LES runs, MG and JM the WRF-GHG runs, AK, AW and PP the EULAG runs, and RH, MPA and CV the ARTM runs. AF, AnF, AR, CK, SK, KG, HB, MG and CG conducted the CoMet campaign in 2018 and processed the measurements used in this study. AnF was PI of the CoMet campaign and contributed to the measurements of DLR. AF and AR performed the DLR-Cessna in-situ measurements and MG and CG the

HALO in-situ measurements. CK and SK performed and analyzed the CHARM-F measurements. JB conducted the FUB-Cessna MAMAP and in-situ measurements. SK performed the analysis of the MAMAP remote sensing data. All authors contributed to the interpretation of the results and to the writing and improvements of the article.

*Competing interests.* No competing interests



*Acknowledgements.* We acknowledge support by the EU project CoCO2 funded through the European Union's Horizon 2020 research

and innovation programme under grant agreement No. 958927 and by the project SMARTCARB funded by the European Space Agency (ESA). We also acknowledge funding for the CoMet campaign by the BMBF (German Federal Ministry of Education and Research) through AIRSPACE (FK 01LK1701B), the State of Bremen, and the Max Planck Society. The work was further supported by the German Research Foundation (DFG) within the Priority Program (SPP 1294) Atmospheric and Earth System Research with the Research Aircraft HALO (High Altitude and Long Range Research Aircraft) under grant BO 1731/1-1. Simulations with COSMO-GHG were supported by a grant from the

Swiss National Supercomputing Centre (CSCS) under project ID s1091 and by the Center for Climate Systems Modelling (C2SM). We thank ECMWF and for providing operational analysis and forecast data as initial and boundary conditions for the high-resolution models.



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

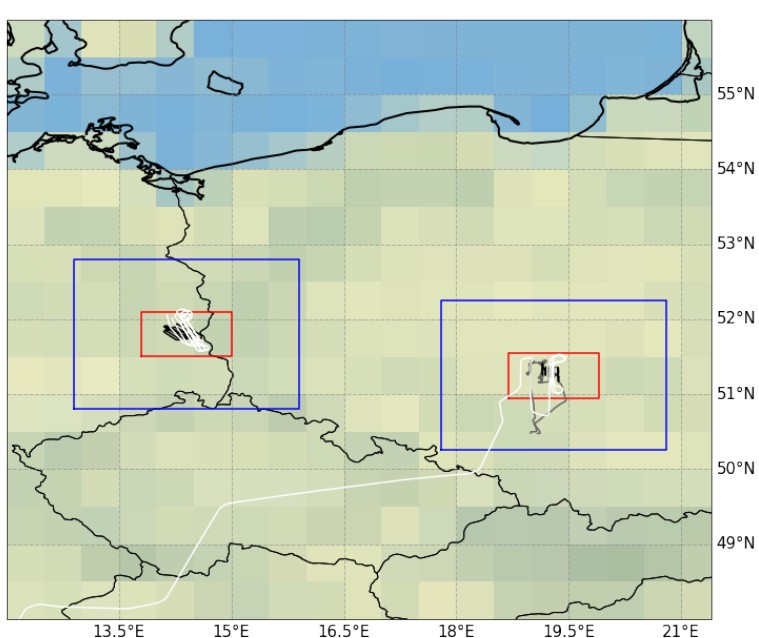

**Figure 1.** Overview of the LARGE (blue) and SMALL (red) model output domains for Jänschwalde (left) and Bełchatów (right). Overlaid are the flight tracks of HALO (white), DLR-Cessna (dark grey, only at Bełchatów) and FUB-Cessna (black) on the corresponding measurement days.





**Figure 2.** Time evolution of the Bełchatów total column $XCO_2$ plume on 7 June 2018 from 05:00 to 13:00 UTC in the NWP models COSMO (top), WRF-GHG (2nd and 3rd row), and in the Lagrangian model ARTM (bottom).





**Figure 3.** Time evolution of the Bełchatów total column XCO$_2$ plume on 7 June 2018 from 05:00 to 13:00 UTC in the LES models WRF-LES (top), ICON-LEM (2nd and 3rd row), and EULAG (bottom).





**Figure 4.** Curtains of the Bełchatów $CO_2$ plume along the DLR-Cessna flight on 7 June 2018.







**Figure 5.** Curtains of potential temperature along the DLR-Cessna flight on 7 June 2018.

**Figure 6.** Curtains of the Jänschwalde $CO_2$ plume along the FUB-Cessna flight on 23 May 2018. Figures for middle release tracer CO2_PP_M.





**Figure 7.** Time series of $CO_2$ along the DLR-Cessna flight at Bełchatów on 7 June 2018. The grey line is the flight altitude (second y-axis).



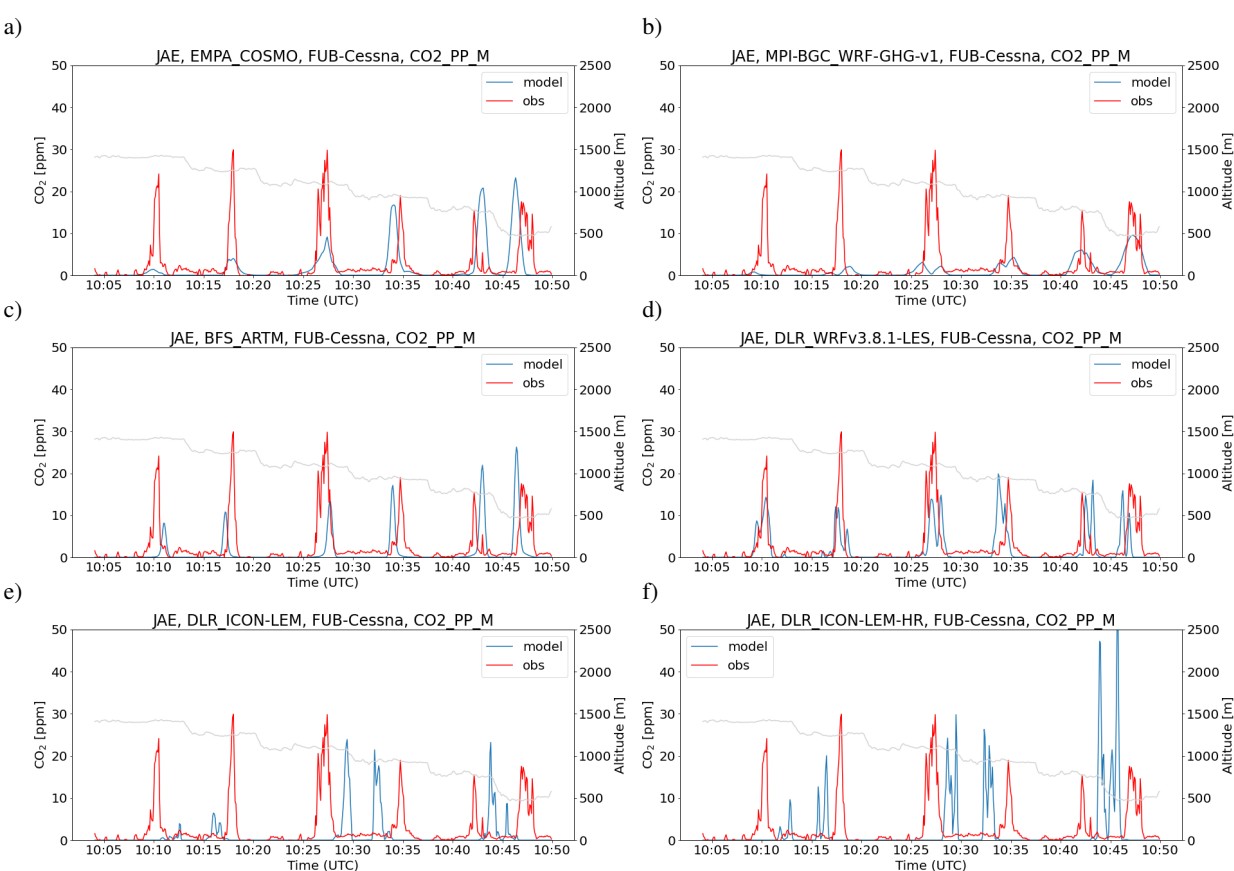

**Figure 8.** Time series of $CO_2$ along the FUB-Cessna flight at Jänschwalde on 23 May 2018. The grey line is the flight altitude (second y-axis)





**Figure 9.** Time series of $CO_2$ column enhancements simulated and observed by MAMAP along the FUB-Cessna flight at Bełchatów on 7 June 2018. The plumes observed around 12:20 and 13:45 UTC, which are not reproduced by any of the models, were measured upwind of the power plant. These plumes are caused by retrieval issues over water surfaces rather by real $CO_2$ enhancements.





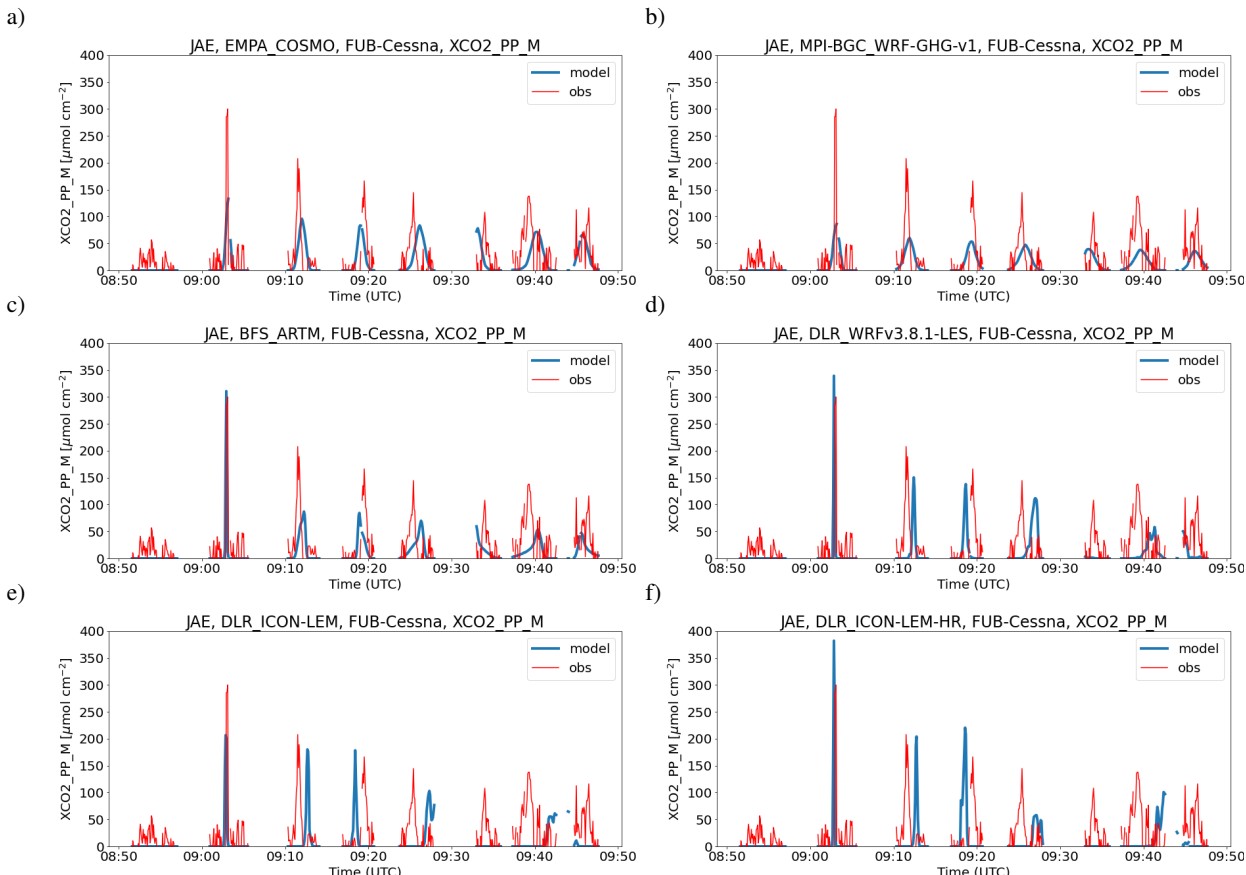

**Figure 10.** Time series of $XCO_2$ along the FUB-Cessna flight at Jänschwalde on 23 May 2018.





a)


b)


c)


d)


e)


f)


g)


h)


**Figure 11.** Time series of $CO_2$ column enhancements simulated and observed by CHARM-F along the HALO flight at Bełchatów on 7 June 2018.





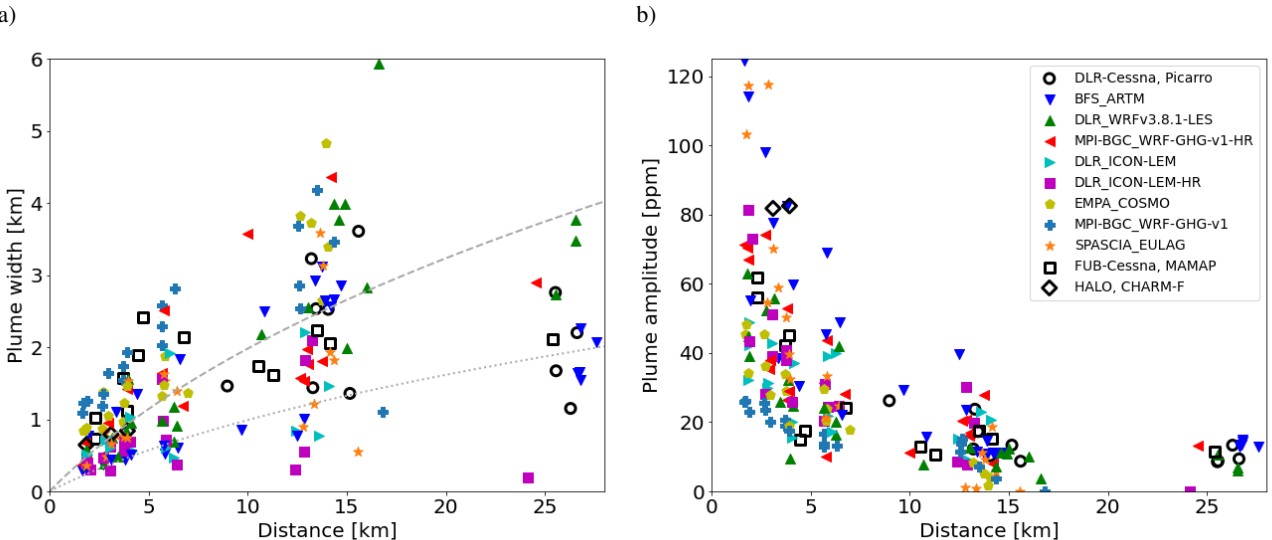

**Figure 12.** Comparison between observed and simulated characteristics of the $CO_2$ plume of the Bełchatów power plant on 7 June 2018 as a function of distance from the source. Plume characteristics were determined by fitting a Gaussian to the individual plume transects. a) Plume widths ($1\sigma$). b) Plume amplitudes. Observations are shown as black open symbols, models as filled colored symbols. Symbols are only shown when the Gaussian fit was sufficiently robust (uncertainty in plume width < 10%) and the plume was not too close to the border of the transect. Grey lines describe plume width of an analytical Gaussian plume model following Briggs (1973) for highly (dashed) and weakly unstable (dotted) conditions.

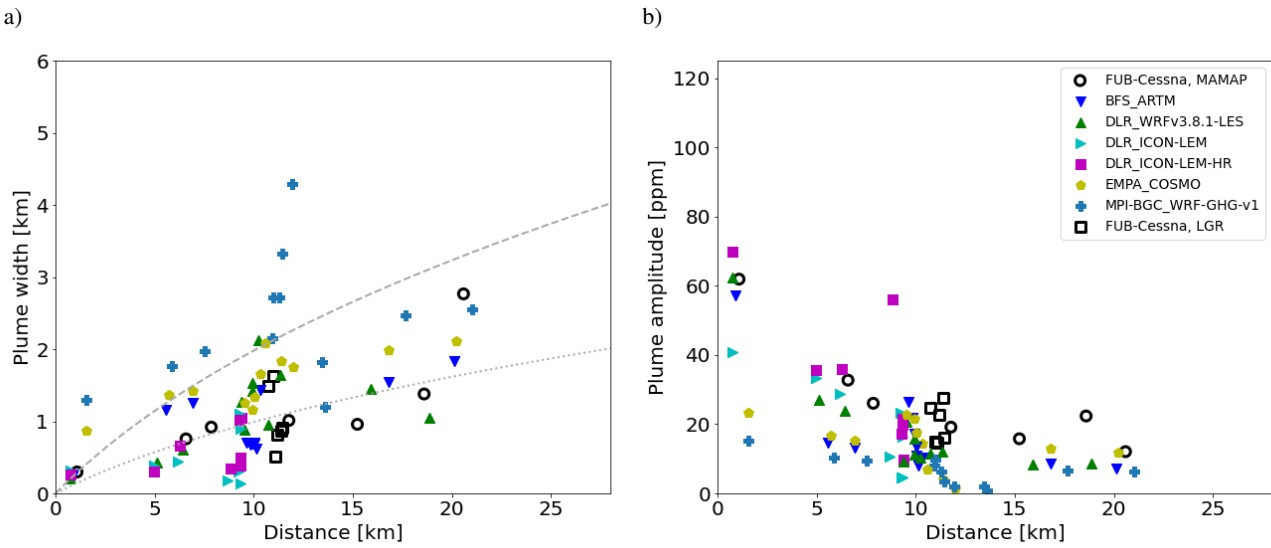

**Figure 13.** Same as Fig. 12 but for the FUB-Cessna measurements collected at the Jänschwalde power plant on 23 May 2018.

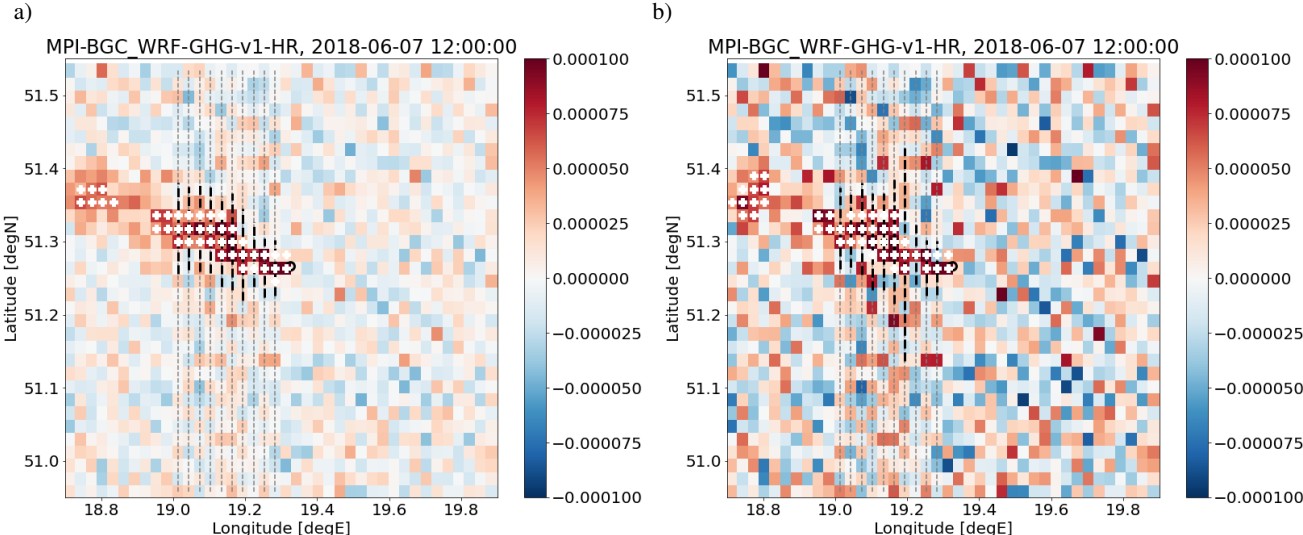

**Figure 14.** Illustration of cross-sectional flux and IME methods for the Bełchatów plume on 07 June 2018 12 UTC as simulated by the WRF-GHG high-resolution (HR) model down-sampled to 2 km x 2 km resolution for (a) low noise (0.5 ppm) and (b) high noise (1 ppm) CO2M instrument noise scenario. For the IME method, pixels above a threshold of $0.4 \, \mathrm{mol \, m^{-2}}$ are marked as white crosses. For the cross-sectional flux method, fluxes through 10 north-south cross-sections (thin grey dashed lines) downwind of the power plant were computed and averaged. The centers and north-south extensions ($\pm 2\sigma$) of the plumes as determined by a Gaussian plume fit are marked with black circles and thick black dashed lines, respectively.



**Figure 15.** Emissions quantified for the Bełchatów plume for all hours of 07 June 2018 by (a,c) cross-sectional flux method and (b,d) Integrated Mass Enhancement method. The upper row is for the low-noise CO2M scenario (0.5 ppm), the lower row for the high-noise scenario (1.0 ppm). The dashed line is the true emissions. Effective wind speeds were obtained as vertically averaged wind speed between 925 and 875 hPa (see text for further details).


**Table 1.** Overview of measurements used for model evaluation. The time range indicated for each aircraft is not the time between take-off and landing but corresponds to the range between start of the first plume transect and the end of the last transect in UTC. The availability of an instrument is indicated by an 'x'.

| Aircraft | Instrument | Parameter | Uncertainty | Reference | Bełchatów 07/06/2018 | Jaenschwalde 23/05/2018 |
|---|---|---|---|---|---|---|
| DLR-Cessna | | | | | 13:19-14:48 | - |
| | Picarro G1301-m | $CO_2$ | 0.15 ppm | Fiehn et al. (2020) | x | - |
| | METPOD | T | 0.15 K | Mallaun et al. (2015) | x | - |
| | METPOD | p | 0.25 hPa | Mallaun et al. (2015) | x | - |
| | METPOD | wind | 0.3 m s$^{-1}$ | Mallaun et al. (2015) | x | - |
| DLR-HALO | | | | | 13:05-13:29 | 08:28-09:33[1] |
| | JIG | $CO_2$ | 0.06 ppm | Gałkowski et al. (2021) | x | x |
| | CHARM-F | DAOD $CO_2$ | 0.8% | Amediek et al. (2017) | x | x |
| | BAHAMAS | T | 0.5 K | | x | x |
| | BAHAMAS | p | 0.3 hPa | | x | x |
| | BAHAMAS | wind | 0.6 m s$^{-1}$ | | x | x |
| FUB-Cessna | | | | | 12:29-14:45 | 08:50-10:50[2] |
| | Los Gatos LGR | $CO_2$ | 0.3 ppm | | - | x |
| | MAMAP | column $CO_2$ enhancement | 0.3% of background | Gerilowski et al. (2011) | x | x |

[1] time of CHARM-F transects, plume not detected by in-situ instrument JIG.

[2] first transects 08:50-10:00 UTC for MAMAP, last transects 10:00-10:50 UTC for in-situ measurements

**Table 2.** Power plants and their $CO_2$ emissions in 2018. Flue gas temperature and effluent flux were estimated from published power plant statistics for Germany (Pregger and Friedrich, 2009) as these were not officially reported.

| Power plant | Longitude [°E] | Latitude [°N] | $CO_2$ emission [Mt yr$^{-1}$] | No. of stacks emitting/total | Stack height [m] | Effluent T [K] | Vol. flux [Nm$^3$ s$^{-1}$] |
|---|---|---|---|---|---|---|---|
| Jänschwalde (DE) | 14.4580 | 51.8361 | 23.1 | 6/9 | 120 | 322 | 790 |
| Bełchatów (PL) | 19.3261 | 51.2660 | 38.4 | 2/2 | 299 | 432 | 330 |





**Table 3.** Overview of participating model systems. The model version is the version of the meteorological core (COSMO, WRF, ICON). The ID is the identifier used to distinguish between different model systems and simulations. The column IC/BC denotes the source of the meteorological data used as initial and boundary conditions. The models WRF-GHG, ICON-LEM and WRF-LES were run in configurations with multiple nests, in which case the column IC/BC describes the initial and boundary conditions for the outermost domain.

| Model | Version | Group | ID | Resolution [km]/levels | Type | IC/BC | References |
|---|---|---|---|---|---|---|---|
| COSMO-GHG | v5.6a | Empa | EMPA_COSMO | 1.1/60 | NWP | COSMO-7 | Brunner et al. (2019) |
| | | | | | | | Jähn et al. (2020) |
| WRF-GHG | v3.9.1.1 | MPI-BGC | MPI-BGC_WRF-GHG-v1 | 2.0/85 | NWP | IFS HRES | Ahmadov et al. (2007) |
| | | | MPI-BGC_WRF-GHG-v1-HR | 0.4/85 | | | Beck et al. (2011) |
| WRF-LES | v3.8.1 | DLR | DLR_WRF-LES | 0.2/56 | LES | IFS HRES | Wolff et al. (2021) |
| ICON-LEM | 2.4.0 | DLR | DLR_ICON-LEM | 0.6/150 | LES | IFS HRES | Kern and Jöckel (2016) |
| | | | DLR_ICON-LEM-HR | 0.3/150 | | | |
| EULAG | - | SPASCIA | SPASCIA_EULAG | 0.2/60 | LES | COSMO-GHG | Prusa et al. (2008) |
| ARTM | 3.0.0 | BFS | BFS_ARTM | 40 levels | LPDM | Observations & COSMO-GHG | Hanfland et al. (2022) |

**Table 4.** Overview of the two model simulations, the minimum time period to be covered, and the longitude/latitude range and resolutions of the two output grids. The SMALL domain corresponds to the minimum domain size to be covered.

| ID | Power plant | Domain | Longitude range [°E] | Latitude range [°N] | Resolution Lon x Lat [°] | Output freq. [hr] | Period Start - End [UTC] |
|---|---|---|---|---|---|---|---|
| BEL | Bełchatów | SMALL | 18.7–19.9 | 50.95–51.55 | 0.003 x 0.002 | 0.25 | 06/06 00:00 - 08/06 00:00 |
| | | LARGE | 17.8–20.8 | 50.25–52.25 | 0.015 x 0.010 | 1.0 | 06/06 00:00 - 08/06 00:00 |
| JAE | Jänschwalde | SMALL | 13.8–15.0 | 51.50–52.10 | 0.003 x 0.002 | 0.25 | 22/05 00:00 – 24/05 00:00 |
| | | LARGE | 12.9–15.9 | 50.80–52.80 | 0.015 x 0.010 | 1.0 | 22/05 00:00 – 24/05 00:00 |



**Table 5.** Emissions from the Bełchatów power plant estimated with the cross-sectional flux (X-Flux) and Integrated Mass Enhancement (IME) methods. Results are presented for 9-15 UTC averaged fluxes from 8 different models as shown in Fig. 15 for a low and a high CO2M measurement noise scenario.

| Method | Noise scenario | Mean [kg s-1] | Bias [kg s-1] | Bias [%] | Std. dev. [kg s-1] | Std. dev. [%] |
|--------|----------|------|------|------|-----------|-----------|
| Truth  |          | 1218 |      |      |           |           |
| X-flux | none     | 1157 | -60  | -5.0 | 200       | 17.3      |
| IME    | none     | 1192 | -26  | -2.1 | 264       | 22.1      |
| X-flux | low      | 1195 | -23  | -1.9 | 224       | 18.7      |
| IME    | low      | 1232 | 14   | 1.1  | 279       | 22.7      |
| X-flux | high     | 1225 | 7    | 0.6  | 275       | 22.5      |
| IME    | high     | 1336 | 118  | 9.7  | 298       | 22.3      |