# Peer review of "Evaluation of simulated CO2 power plant plumes from six high-resolution atmospheric transport models"

_Atmospheric Chemistry and Physics, 2022_

## Author Comment (AC1)

We would like to thank Anna Karion for the careful reading and valuable comments, which we try to address point-by-point. In the following, reviewer comments are in italics, answers in normal letters and suggested new text in blue color.

*Overall, this is a very nice paper: a well-thought-out experiment was constructed to compare many atmospheric transport and dispersion models under the condition of known emissions. Presentation is very good, with clear nice figures and a lot of good additional information in the SI; it also reads well without getting too long (!) and losing the interest of the reader. I have comments and suggestions below, and have tried to note when a comment is just an optional suggestion to improve the analysis or a question that arose in my mind as I read it, especially the last comment below.*

*[general note: the authors should cite Angevine et al., https://doi.org/10.5194/acp-20-11855-2020] - it's different but similar in that they simulated powerplant plumes that had known emission rates.*

Thank you for pointing us to this relevant publication. We added the following sentence at line 73:

Another related study was conducted by Angevine et al. (2020), who simulated the plume of a power plant with a single Lagrangian transport model to analyze different sources of error in top-down emission estimates including uncertainties in winds and boundary layer heights as also addressed in our study.

*Specific comments:*

*L 18 this raises the question: how did the LPDM perform?*

We added the following sentence to the abstract:

The Lagrangian model, which was the only model driven by winds observed from the aircraft, quite accurately captured the location of the plumes but generally underestimated their width.

*L114 & 130 Perhaps "Table" should be spelled out here as it is in Line 97? Up to the editors.*

We replaced Table by Tab. on line 97 to be consistent.

*Fig. 1, what is the underlying pixelated color? land area perhaps? elevation? Does the pixel size here mean anything? (~0.5 degrees?) relative to model resolution? (I imagine not as the models are all finer-scale).*

The underlying map has no meaning other than guiding the eyes. We added the following sentence to the figure caption:

The background map shows the contrast between land (green shadings) and sea areas (blue shadings).

*Lines 150-170: Can the authors explicitly state whether these two versions of WRF are the same in the outer domains? (same configuration?). I see a lot of specifics on WRF-GHG configuration, and different specifics on the WRF-LES configuration. The domains are different resolutions (10 km - 2km nest vs. 5-1-0.2), so I am guessing these were different. But the first*

*description does not include the version, and the second does not include the land surface, cumulus parametrization, radiation, etc.... so hard to compare! It would be good to know how similar the meso-scale portion of WRF-LES is to the WRF-GHG run to see if any differences are due to the outer domain or due to the LES.*

The two versions were set up independently with different domains and configurations. Additional information on model versions and configurations and the different model domains will be provided in a new supplement S5 to make clear which elements are shared and which are not. WRF-GHG was run using WRF version 3.9.1.1 for 3 nested domains at 10 km, 2 km and 400 m resolution, the out domain covering whole Europe. WRF-LES was also run using WRF version 3.9.1.1 for Belchatow but using 3.8.1 for Jänschwalde. WRF-LES simulations were performed for three nested domains at 5 km, 1 km and 200 m resolution, the outer domain covering a portion of central Europe (see Figure 1). Unlike WRF-GHG, WRF-LES does not make any use of the Chem addons of WRF-Chem. The LES mode was only applied in the innermost domain. The main differences in terms of parameterizations are the microphysics scheme (WRF-GHG with single moment 5-class scheme, WRF-LES with Morrison 2-moment scheme) and the land surface scheme (WRF-GHG with CLSM v4 and WRF-LES with Unified Noah LSM). Both models used, however, the same boundary layer scheme (MYNN 2.5). Details on the chosen parameterizations for all model domains will be provided in the supplement. The description of the two models in the manuscript will be updated.

[Figure]

**Figure 1**: Simulation domains of WRF-GHG (a), WRF-LES Belchatow (b) and WRF-LES Jänschwalde (c). The figure will also be provided in Supplement S5.

*Eq 1 and lines 253-260. The authors should explain what is cp(y) and its units? A is called a scaling constant, then an area integral but then it is in ppm. If A is an integral shouldn't it have some kind of other units like ppm * distance? (sigma has units of distance also right? if it is the same sigma in both parts of the equation)? Also somewhere the authors should note that "CO2" here is a mole fraction, i.e. in units of micromoles per mole of (dry?) air, or ppm. Is it a mole fraction, not a concentration, right?*

We are sorry for having been sloppy and even incorrect in the case of the area integral $A$. The quantity $c_p$ is either the $CO_2$ mole fraction in units of ppm (for the comparison with in-situ measurements) or the column integral of $CO_2$ in units of mol cm$^{-2}$ (for remote sensing data). Accordingly, the units of the area integral $A$ are either ppm * m or mol cm$^{-2}$ * m (=100 mol cm$^{-1}$), since the plume width sigma has units of m. The text will be changed accordingly.

*L265 What is a YAML file?*

YAML files are self-descriptive text files for storing data structures following the YAML (Yet Another Markup Language) specifications. This information will be added.

*Figs 2 & 3 these are very nice figures for showing qualitative model differences at a glance.*

Thank you.

*L311: Was the same constant value chosen based on the best fit to the models? Also, did a constant value fit all transects? I.e. there was not temporal evolution of the background assumed during the flight? (or altitude dependence)? [I see the altitude dependence is mentioned later in L317-319]*

Yes, the same constant background was chosen for all flights for this qualitative analysis. A clear altitude dependence was only visible at the transition between the ABL and the free troposphere but not within the ABL. For the quantitative analysis, however, where Gaussian curves were fitted to all plume transects, we did not assume a constant background. A linearly changing background was computed separately for each transect and subtracted before fitting the Gaussian. The linearly changing background was computed as a line connecting the 10% percentile of the first 1/5 of data points with the 10% percentile of the last 1/5 of data points in the transect. We are sorry that this information was missing. It will be added to the description of the Gaussian fit procedure in Section 2.3.

*P12: this discussion makes me wonder if turbulence profiles (TKE for example, or just sigma_w) were compared between models (assuming measurements were not available on the aircraft). Even just comparing between the models might be interesting to understand the mixing in addition to Potential temp. (Just a suggestion)*

We agree that this would be an interesting analysis, but unfortunately neither TKE nor vertical winds were stored by the models as the variables were not included in the protocol. Note that an analysis of TKE would not be straightforward. As we recently showed in a study on Lagrangian simulations at the kilometer-scale (Katharopoulos et al., 2022, https://doi.org/10.1007/s10546-022-00728-3), part of the turbulence spectrum is already resolved by the grid-scale winds in weather prediction models operating at resolutions of the order of 1 km. TKE values (from a prognostic scheme of subgrid-scale turbulence) can therefore not easily be compared between models operating at different resolutions near the grey zone of turbulence.

*I may have missed this further up, but were the emissions for the models the same listed in Table 2, i.e. the annual average? Is there temporal variability in these emissions? perhaps the models were using emissions reported at the hourly level (here in the US we get hourly Continuous Emissions Monitoring System data from the stack measurements for Powerplants)? We have found hugely varying emissions in time even from hour to hour, and even in predominantly coal plants here in the US. Presumably that is not the case for these two, and emissions are constant?*

Yes, all models used the same emission rate corresponding to the annual mean value. Because $CO_2$ was treated as a passive tracer, a different emission rate simply corresponds to a differently scaled $CO_2$ concentration field.

In response to your question and considering the recent publication by Nassar et al. (2022), which demonstrated that actual $CO_2$ emissions at Belchatow can be readily estimated from actual power generation, we had a closer look at the power generation and corresponding emissions at the two power plants during the observation days. The figure below shows the power generation at Belchatow on 7 June 2018 and at Jänschwalde on 23 May 2018. The approximate time periods covered by the aircraft observations are highlighted. At Belchatow, power generation was reduced at night and increased in the early morning hours to a rather constant value for the rest of the day. At Jänschwalde, in contrast, power generation was reduced during daytime between 10 and 17 CEST. By comparing the power generation during the observations periods to the annual mean power generation, we estimate that $CO_2$ emissions were 23% higher than the annual mean during the observations at Belchatow and even 28% higher at Jänschwalde. We added this information to Table 2, rescaled all model simulated $CO_2$ fields (by factor 1.23 for Belchatow and 1.28 for Jänschwalde), and regenerated all corresponding figures. Overall, the scaled simulations fit the observations better.

[Figure]

**Figure 1**: Hourly power generation at the two power plants during the observation days. The average energy production during the observations (points enclosed by the blue and orange boxes, with points denoting the end of a 1 hour averaging interval) were used to scale the simulated $CO_2$ concentrations.

We will add this information to Section 2.2 (Modelling protocol) as follows:

Constant emission rates corresponding to the annual means reported to E-PRTR for the year 2018 were used in all simulations (Table 2). However, the actual emission rates during the observation periods were different. Following Nassar et al. (2022), we estimated hourly $CO_2$ emissions by comparing actual energy production during the observation periods with annual mean energy production by the two power plants. We assume that the period of power generation relevant for the observations at Belchatow was 7 June 12:00-14:00 UTC. For the observations at Jänschwalde, the corresponding period was 23 May 08:00-10:00 UTC. Based on these assumptions, we estimate that the actual $CO_2$ emission rate at Belchatow was 47.4 Mt yr$^{-1}$, i.e.

23% higher than the annual mean. At Jänschwalde, the actual emission rate was 29.5 Mt yr$^{-1}$, i.e. 28% higher than the annual mean. Details of the computation including tables of annual and hourly energy production and references to the data sources are provided in Supplement S4. To account for the higher $CO_2$ emission rates during the observation periods, all simulated $CO_2$ fields were scaled by a factor 1.23 for Belchatow and by a factor 1.28 for Jänschwalde.

*Fig 12 discussion: It would make things clearer if these parameters (plume width and amplitude) were referred back to the equation where they are defined, with those symbols? (for example, I woudld think that sigma was the plume width although that's not the definition in the methods section near the equation?).*

We agree. Although the discussion of Fig. 12 in Section 3.4 mentions that "plume widths were geometrically corrected to represent the width perpendicular to the plume axis", the caption of Fig. 12 only refers to $\sigma$, but what is shown is actually the product of $\sigma$ and *cf*. The caption will be changed accordingly. Furthermore, we changed the sentence at line 427 as follows:

Width ($\sigma$ *cf*), amplitude (maximum), and integral area ($A$) of the fitted Gaussian were determined from ..

The computation of the amplitude was already described in Sect. 2.3 as "Plume amplitude was computed as the maximum of the Gaussian curve at the location $y = \mu$". Since the first sentence in Sect. 3.4 refers back to Sect. 2.3, we don't think it is necessary to repeat this.

*Fig 12-13. (Suggestion/Comment) I realize the point here is to compare the evolution of the plume with distance for the various models, but it might be interesting to consider the integral under the plume for each model vs. observed for all the transects/distances. Presumably, this should be the same at the different distances (well only if wind & PBL are constant - so maybe not). (is this A in the equation? I'm not sure). IF the integral is not the same with distance then it points to changes in the wind, PBL, plume separation (some of it going above the ABL and advected faster), etc. that might explain deviations from the gaussian plume model.*

We followed your suggestion and in addition to plume width and amplitude also computed the integral under the plumes. A new panel (c) was added to Figures 12 and 13 presenting these results. We also compared the mean plume integral averaged over all transects between models and observations and found an excellent agreement, which indicates that the $CO_2$ emissions assumed in the simulations are consistent with the observations. Please see our responses to the other reviewer for the additional text and figures added to the manuscript.

*Fig 14, very nice figures, and easy to understand the symbols etc. What are the units on the colorbar?*

The units are mol cm$^{-2}$. The information will be added to the figure caption.

*L504. Could the authors comment on how one would determine the wind if one was using the satellite data to determine the flux using the cross-sectional method? I.e. one would perhaps use modeled winds in the same way as was done here. But given the current "image" data is generated by the same model as is being sampled for wind, it's sort of a best (perfect)-case scenario. What if the wind from a one model was used to determine emissions from the image of a different model? That may be outside the scope of this paper, but it's relevant in determining the ability to estimate accurate emissions if the wind model is not exactly like reality,*

*as it is in the model world here. Perhaps given the range of these wind values (the average used in the emissions estimate) from the different models, one could arrive at an additional uncertainty from this component of the analysis? (I see later the wind speed issue is addressed for the IME method).*

The estimation of the effective wind speed of the plume is a critical issue for both methods. The reviewer is right that we are only investigating a perfect-case scenario, where the wind fields are perfectly known. Our goal is to demonstrate that even in this ideal case important uncertainties remain due to the turbulent nature of the plumes and due to the fact that in practice we will not know exactly at what altitude the plume is located. To make this clearer we will add the following sentences to the first paragraph of Section 3.5:

In order to translate vertical columns into fluxes, both methods require the estimation of an effective wind speed (or transport speed) of the plume. Here, it is determined separately for each model using the respective 3D model wind fields. In case of real satellite observations, however, the transport speed would be estimated from a meteorological analysis (see e.g. Nassar et al., 2017, 2021, 2022), which comes with an additional uncertainty because the analyzed winds will be different from reality. Although the wind fields are perfectly known in our case, we will show that the estimation of the effective wind speed is affected by uncertainties due to turbulent wind fluctuations and due to the fact that it is not known exactly at what altitude the plume is located.

The question of the uncertainty in analyzed winds is an interesting and important one, but we think it is outside the scope of our study. Nassar et al. (2022; https://doi.org/10.3389/frsen.2022.1028240) addressed the question to some extent by using winds from two different meteorological analyses, i.e. MERRA-2 and ERA5.

*L545-547 This is a very important point, in a current climate where satellite observations are meant to solve all our emission quantification problems -- we will still be relying on a model to do this, and the model still requires an accurate estimate of wind speed in the PBL and PBL depth.*

Yes, indeed. We think that more studies on the uncertainty of wind fields in meteorological analyses are needed, not only focusing on uncertainties at the surface but over the whole depth of the PBL.

*[overall comments / musings, no need to address in the paper unless it would be easy to do!]:*

*I wonder if this set of very valuable model simulations could be used to understand how important dispersion and turbulence are for accurately simulating tracer concentrations (mole fractions) relative to the importance of the underlying mean meteorology (mean wind speed and PBL depth for example).*

*I.e., often when estimating emissions using a model, the underlying meteorology (not the dispersion) is usually evaluated: wind speed, wind direction, and PBL. If the underlying met has no bias in these quantities, the transport is considered "validated". Unfortunately, it is not clear whether there is a way to evaluate dispersion as well? I would guess not without observations of TKE or such a quantity. I wonder if the authors could comment on the ability of the models they investigated to simulate mole fractions at a given location even when performing well for wind speed, direction & PBL depth. It might be nice (if the authors agree with this*

*based on their findings) to emphasize this issue of turbulence, mixing, and dispersion as important separately from wind speed, direction and ABL depth, which are obviously crucial but not the whole story.*

Capturing the mean meteorological conditions (wind direction, speed and PBL depth) is obviously an important first step as shown by the ICON-LEM model, which simulated a wrong wind direction and too high wind speeds and therefore showed comparatively poor agreement with the observations. ABL depth is essential for the comparison with in-situ observations, but less so for the comparison with total columns as demonstrated by EULAG, which significantly underestimated PBL depth but showed good performance when compared with vertical column observations.

Dispersion is indeed much more difficult to validate. The dispersion in Lagrangian dispersion models has typically been tuned based on tracer release experiments, but the same is not true for Eulerian models such as COSMO, WRF or ICON, which have primarily been developed for weather prediction or climate applications. Whether the dispersion of a tracer is well represented has typically been of secondary importance, but this is gradually changing since weather centers (such as ECMWF or DWD) are increasingly engaged in atmospheric transport and inverse modelling. When comparing total columns between observations and models, dispersion is arguably less relevant compared to mean winds, because the comparison can be performed in terms of plume (or line) integrals, which are only little affected by errors in dispersion. An accurate representation of plume dispersion, however, is important for Observing System Simulation Experiments assessing the performance of future satellite missions as done e.g. by Kuhlmann et al. (2021) using the COSMO-GHG model. The too strong dispersion in this model likely leads to an underestimation of the number of cases, in which a plume of a city or a power plant is detectable against instrument noise and background variations. It is thus difficult to provide a general answer to this question, since the answer will depend on the specific application.

---

## Author Comment (AC2)

We would like to thank Ray Nassar for the careful reading and valuable comments that we try to address point-by-point in the following. Reviewer comments are in italics, answers in normal letters and suggested new text in blue color.

*"Evaluation of simulated $CO_2$ power plant plumes from six high resolution atmospheric transport models" by Brunner et al. assesses model simulation capabilities to understand their applicability for use in power plant emission estimation from aircraft and satellite observations. The study uses real aircraft in situ and remote sensing observations from the CoMet aircraft campaign of 2018 over the Belchatow and Jänschwalde power plants and simulated CO2M observations (with 2 noise levels) with six models including Eulerian, Lagrangian and Large Eddy Simulation (LES) models. This is a useful and informative study that enhances our understanding of model capabilities and limitations, but the authors correctly warn that a complete comparative evaluation of the models cannot be carried out based on just a few overflights. The quality of the study is high and the interpretation of the results is generally sound, as the authors are careful not to over-interpret the relatively small number of examples used in the study.*

*I would have liked the study to go the additional step of actually reporting the estimated emissions of the two power plants based on the real aircraft observations and not only the simulated CO2M data (Table 5). The spread in derived emission estimates that would be obtained with the different models would be informative, although not essential for this study.*

Indeed, this additional step could have been taken but we wanted to focus on the model simulations and leave such an analysis to the measurement groups. Nevertheless, we took a step towards a more quantitative analysis. As described in more detail in our response to the other reviewer, we estimated the actual $CO_2$ emissions during the observation periods from actual energy production rates following your publication (Nassar et al., 2022). Based on this analysis, we scaled all simulated $CO_2$ fields by a factor 1.23 for Bełchatów and 1.28 for Jänschwalde (actual energy production was higher compared to the annual mean) and regenerated all figures with $CO_2$ data. The scaled model results agree better with the observations. Furthermore, in Sect. 3.4 (Statistical properties of the plume), we added an analysis of plume integrals to the analysis of plume widths and amplitudes and compared the model results with the observations. This comparison shows that the scaled model results are fully consistent with the observations whereas using annual mean $CO_2$ emissions would not have been consistent. The two new figures are shown below

a)                                            b)

[Figure]

Figure 1: Plume integrals [ppm km] for (a) Bełchatów and (b) Jänschwalde. See Figure 12 and 13 in manuscript for a legend of symbols.

The following paragraphs will be added in Sect. 3.4:

Discussion of results for Bełchatów:

The plume integrals (i.e, the areas under the Gaussian curves) presented in Fig. 12c correspond to the integrated amount of $CO_2$ along each transect in units of ppm km. Since $CO_2$ is transported as a passive gas, they are expected to stay constant with distance unless (i) the wind speed or wind direction changes with distance (or with time since the transects were flown at different times), (ii) the plume extent is not fully covered by all transects, or (iii) the plume is not yet homogeneously mixed over the full depth of the ABL, such that a mole fraction measured by an in-situ instrument at a given altitude is not representative for the ABL column mean. The figure suggests that the plume integrals are indeed not constant but decrease with distance, more clearly in the measurements than the simulations. The reason for this could be any combination of the above possibilities. The integrals also enable a quantitative comparison between observations and models. The mean (and standard error of the mean) averaged over all models (excluding ICON-LEM due to its too high wind speeds and excluding points with unrealistically low values below 10 ppm km) and over all distances is 105.6±2.8 ppm km (n=126). The corresponding mean over all observations is 111.9±11.1 ppm km (n=26). The two values agree within their combined uncertainties suggesting that the simulations are consistent with the observations.

Discussion of results for Jänschwalde:

Different from Bełchatów, the plume integrals remain approximately constant with distance. The mean averaged over all models except ICON-LEM is 55.5±2.1 ppm km (n=92) and the corresponding mean over all observations is 57.0±5.6 ppm km (n=13). Again, the two values agree within their combined uncertainties. Using annual mean instead of actual $CO_2$ emission rates in the simulations would have resulted in too low plume integrals inconsistent with the observations for both Bełchatów and Jänschwalde. This finding agrees with a recent study by Nassar et al. (2022), who demonstrated that it is necessary to account for actual power generation to explain day-to-day variations in $CO_2$ emissions from Bełchatów estimated from individual OCO-2 and OCO-3 satellite overpasses.

In the conclusions section, the following sentences will be added to the first paragraph:

The $CO_2$ emissions assumed in the simulations correspond to values officially reported for the year 2018 but scaled by a factor 1.23 for Bełchatów and 1.28 for Jänschwalde to account for the fact that hourly energy production rates were higher during the observations than annual mean production rates. The amount of $CO_2$ integrated along individual plume transects was highly consistent between simulations and observations when the emissions were scaled in this way.

*With or without the above suggestion, I only recommend some minor specific changes before I would deem the study acceptable for publication in Atmospheric Chemistry and Physics.*

**Specific Points**

*Line 1: "dominated" is too strong of a term, since there are other major sources like urban $CO_2$ emissions from which a large fraction is from transportation or residential heating, rather than facilities. Based on the introduction, the contribution from facilities is about 58%.*

"Dominated" is indeed to strong. The sentence will be changed to

Power plants and large industrial facilities contribute more than half of global anthropogenic $CO_2$ emissions.

*Line 10: The description should clarify "NWP models extended for atmospheric tracer transport" or something like this rather than just calling them NWP models.*

Thank you, this will be changed as suggested.

*Line 36: The actual prevalence of stack monitors is somewhat uncertain. Recommend changing "often measured" to "sometimes measured".*

In Europe and probably other developed countries such measurements are demanded by regulation, but it is unclear what fraction of industrial and power plants worldwide have such a system. We agree that "sometimes" may be a better description.

*Line 46: Recommend including OCO-3 (Nassar et al., 2022 https://www.frontiersin.org/articles/10.3389/frsen.2022.1028240/full), which has enhanced capabilities relative to OCO-2 for locations of interest within the latitude range covered (up to ~52°N), and furthermore is highly relevant to the Bełchatów examples.*

Thank you for pointing at this publication, which is clearly relevant in the context of our study. We added a reference on line 46 and also added references in Sections 2.2 (Modelling protocol), 3.4 (Evaluation of plume statistics) and 3.5 (Emission quantification with a CO2M like satellite). Note that in response to a question of the second reviewer and considering your recent publication (Nassar et al. 2022), we estimated the actual emissions from Bełchatów and Jänschwalde by comparing hourly energy production data during the observations with annual mean energy production. All simulated $CO_2$ concentrations were scaled accordingly (with a factor of 1.23 for Bełchatów and 1.28 for Jänschwalde) and all corresponding figures were updated.

*Lines 52-56: Should really cite Zheng et al. (2019) as an example for 3D atmospheric transport modelling for $CO_2$ point source emission estimates https://iopscience.iop.org/article/10.1088/1748-9326/ab25ae, which demonstrated the exact problem with small errors in the wind direction.*

We added a reference to Zheng et al. (2019) on line 56.

*Line 115: The authors made an appropriate decision to optimize the location used in approximating the power plant as a point source based on actual stack locations.*

This was indeed an important consideration, which led to clearly improved results.

*Line 277: The authors need to double-check the stated sunrise time of 3:33 CET on June 6, which seems too early, as they are likely reporting the onset of twilight rather than the actual sunrise. I am unsure however which one (onset of twilight or actual sunrise) is more relevant for the ABL height. At minimum, they need to be more careful with wording.*

Thanks a lot for spotting this error! According to timeanddate.com, the sunrise (not twilight) was almost one hour later at 4:28 CET. We will change the text accordingly.

*Figures 2-4: The comparisons are very interesting and informative regarding the model spread.*

Thank you.

*Figure 4: Variation in vertical dimension between the models strongly suggests that this will be less of an issue with satellite column data and this is confirmed by comparisons is 3.3 (Fig 9 – 10).*

Yes, absolutely. This important point is therefore mentioned also in the conclusions.

*Figures 2-11: If the authors can make the model name label on the Figures more prominent in comparison to other text, it would significantly improve clarity for the reader.*

We increased the label sizes and placed the model names as the first item in all figure titles.

*Figure 14: Units should be specified for colour scale of the figure.*

The units are mol cm$^{-2}$. The corresponding information will be added to the figure caption.

*Finally, the conclusion ends somewhat abruptly. I think the manuscript would benefit with an additional paragraph dealing with the bigger picture, where the authors put this study in the context of the expected capabilities and limitations of power plant $CO_2$ emissions monitoring, verification and support (MVS) with CO2M.*

We agree that it would be good to add a few lines placing our study in context. We added the following sentences at the end of the conclusions section:

A potentially important application of high-resolution model simulations as performed in this study is the estimation of point source emissions from satellite observations through inverse modelling. However, accurately simulating the location and structure of the corresponding plumes will remain a challenge especially in the presence of turbulence. Simple Bayesian inversions where simulations and observations are compared locally on a pixel-by-pixel therefore seem little suited but more advanced methods, e.g. using non-local metrics as proposed by Vanderbecken et al. (2022), will be necessary. Whether such methods can outperform simpler methods such as Gaussian plume matching and mass balance approaches that do not require any expensive model simulations, will have to be seen.

High-resolution simulations are invaluable, however, for testing the capabilities of future satellites or other measurement platforms in Observing System Simulation Experiments as shown e.g. by Kuhlmann et al. (2019,2021). Our study shows that simulations performed at a resolution of 1 km or better are able to provide a highly realistic representation of real plumes.

Furthermore, we added the following lines to the paragraph discussing the results from the analysis of synthetic CO2M data.

Our estimate of a 20% uncertainty is higher than the average value of about 12% recently estimated by Nassar et al. (2022) for single Snapshot Area Mapping (SAM) images from the OCO-3 satellite over Bełchatów. They acknowledge that their value could be an underestimate of the total uncertainty, but on the other hand, it was consistent with absolute differences

between estimated and expected (from actual power generation) emissions. A 20% uncertainty may thus be a conservative estimate.